# Spindle motility skews division site determination during asymmetric cell division in Physcomitrella

Elena Kozgunova [1,2,3✉], Mari W. Yoshida[3], Ralf Reski [1,4,5] & Gohta Goshima [3,6✉]

Asymmetric cell division (ACD) underlies the development of multicellular organisms. In animal ACD, the cell division site is determined by active spindle-positioning mechanisms. In contrast, it is considered that the division site in plants is determined prior to mitosis by the microtubule-actin belt known as the preprophase band (PPB) and that the localization of the mitotic spindle is typically static and does not govern the division plane. However, in some plant species, ACD occurs in the absence of PPB. Here, we isolate a hypomorphic mutant of the conserved microtubule-associated protein TPX2 in the moss *Physcomitrium patens* (Physcomitrella) and observe spindle motility during PPB-independent cell division. This defect compromises the position of the division site and produces inverted daughter cell sizes in the first ACD of gametophore (leafy shoot) development. The phenotype is rescued by restoring endogenous TPX2 function and, unexpectedly, by depolymerizing actin filaments. Thus, we identify an active spindle-positioning mechanism that, reminiscent of acentrosomal ACD in animals, involves microtubules and actin filaments, and sets the division site in plants.

---

[1] Plant Biotechnology, Faculty of Biology, University of Freiburg, Freiburg 79104, Germany. [2] Institute for Advanced Research, Nagoya University, Furo-cho, Chikusa-ku, Nagoya 464-8601, Japan. [3] Division of Biological Science, Graduate School of Science, Nagoya University, Furo-cho, Chikusa-ku, Nagoya, Aichi 464-8602, Japan. [4] CIBSS – Centre for Integrative Biological Signalling Studies, University of Freiburg, Freiburg 79104, Germany. [5] Cluster of Excellence livMatS @ FIT – Freiburg Center for Interactive Materials and Bioinspired Technologies, University of Freiburg, Freiburg 79110, Germany. [6] Sugashima Marine Biological Laboratory, Graduate School of Science, Nagoya University, Sugashima, 429-63, Toba 517-0004, Japan. ✉email: kozgunova@gmail.com; goshima@bio.nagoya-u.ac.jp

Chromosome segregation during mitosis and meiosis is driven by a complex microtubule (MT)-based apparatus known as the spindle. Animal spindles are known to be mobile and their final position corresponds to the future cytokinesis site, which in turn could determine daughter cell fate after asymmetric division. The mechanism by which the spindle is positioned and spatially controls the assembly of the cytokinetic machinery has been well studied in animals, and the critical roles of force-generating machinery, such as dynamic MTs, actin, and motor proteins, have been elucidated[1,2]. However, in plants, it is believed that the preprophase band (PPB), a plant-specific MT-actin belt formed prior to mitotic entry, determines the future cell division site[3–5]. The mitotic spindle always forms perpendicular to and at the site of the PPB, perhaps by the action of bridging MTs that connect the spindle to the former PPB site[6].

The spindle position in plant cells is considered to be static, unless a strong force (1600–3350 × g) is applied through centrifugation, which also causes other cytoplasmic components to translocate[7,8]. The static nature of spindles is consistent with the fact that plants lack centrosomes, which play key roles in spindle translocation in animal somatic cells[1,2]. Multiple proteins co-localized to the PPB are required to establish and maintain the cortical division zone (CDZ), towards which the cytoskeleton-based cytokinetic machinery, known as the phragmoplast, expands while recruiting cell plate components[9,10]. However, the essential role of PPB in determining the cell division site has been challenged by a study that characterized cell divisions in the roots of Arabidopsis thaliana trm678 mutants lacking PPB. The findings showed that PPB is not essential for division plane determination[11]. In addition, global tissue organization and plant development, although more variable, were comparable to wild-type plants, implying that ACDs are executable without PPBs.

The moss Physcomitrella (Physcomitrium patens, formerly called Physcomitrella patens) is an attractive model plant for studying PPB-independent division plane determination, as most cell types naturally lack PPBs, but are capable of oriented cell division and patterning into complex 3D structures, such as gametophores (leafy shoots)[12,13]. We have previously shown that the MT structure, called the gametosome, appears in the cytoplasm transiently at prophase and acts as a determinant of spindle orientation[13]. However, gametosomes are dispensable for spindle MT generation and spindle positioning.

In animal cells, the mitotic spindle is assembled through rapid MT nucleation and amplification aided by multiple proteins, including γ-tubulin, augmin, and TPX2 (targeting factor for Xklp2)[14]. A previous study in A. thaliana, using a combination of knockout and cross-species antibody injection, suggested that TPX2 is an essential gene[15]. However, these results were recently questioned when several viable AtTPX2 t-DNA insertion mutants were obtained[16]. In addition to canonical TPX2, several TPX2-like genes lacking one or more functional domains have been identified in A. thaliana. Among them, TPX2L3 lacks a C-terminal kinesin-binding motif but is strongly associated with Aurora kinases and is essential for embryogenesis[16]. However, the mechanism by which TPX2 contributes to spindle formation and MT amplification in plant cells remains unknown.

In this study, we aimed to characterize TPX2 function in the spindle assembly of P. patens, wherein many research tools, including inducible RNAi, endogenous gene tagging, and highly efficient CRISPR, are easily applicable[17,18]. In addition to TPX2's role in MT amplification during early mitosis, we found an unexpected function of TPX2 in maintaining spindle position during asymmetric cell division in gametophores.

## Results

### P. patens TPX2 phylogeny and localization during mitosis.
We identified five genes homologous to TPX2 in the P. patens genome using a BLAST search and named them TPX2-1 to -5 (Fig. 1a). TPX2-1 to -4 proteins showed higher similarity to canonical TPX2 in seed plants (e.g., A. thaliana and Oryza sativa), whereas TPX2-5 appears to have lost the N-terminal Aurora-binding motif[16,19,20], but retained the highly conserved C-terminal domains and, to a certain extent, γ-tubulin activation motifs[21] (Fig. 1b).

To observe the localization of TPX2s in the cell, we fused fluorophores (Citrine (Cit), mNeonGreen (NG), or SunTag (ST))

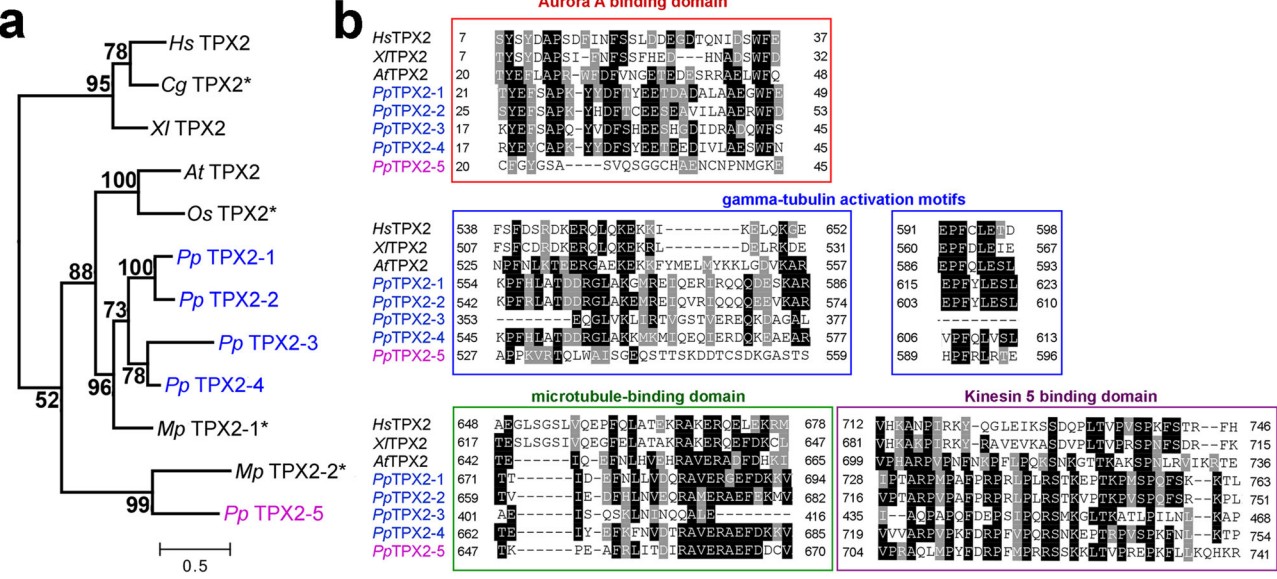

**Fig. 1 TPX2 homologs in P. patens. a** Phylogenetic analysis revealed two distinct groups of TPX2 proteins in P. patens: Pp TPX2-1 to −4 (blue), which are more similar to TPX2 genes from seed plants, and atypical TPX2-5 (magenta). Asterisks mark predicted proteins, numbers show bootstrap values. Bar, 0.5 amino acid substitutions per site. Note that AtTPX2L3 and AtTPX2L2 were not added to this tree, since they lack the C-terminal region that is conserved in canonical TPX2 proteins. Hs: Homo sapiens, Gg: Gallus gallus, Xl: Xenopus laevis, At: Arabidopsis thaliana, Os: Oryza sativa, Pp: Physcomitrium patens, Mp: Marchantia polymorpha. **b** Alignment of TPX2 proteins. Conserved residues are boxed, whereas similar amino acids are hatched.

in-frame at the C- or N-terminus of each endogenous *TPX2* gene (Fig. 2). We did not observe any fluorescence from NG-TPX2-3, which is consistent with the PEATmoss database, indicating extremely low expression of *TPX2-3* in protonema (tissue comprised of frequently dividing tip-growing cells) and gametophores[22]. Various levels of fluorescence were detected for the other TPX2 proteins. In prophase, prior to nuclear envelope breakdown (NEBD), endogenous TPX2-1, −2, and −4 localized to the MT cap formed at the apical side of the nucleus (Fig. 2a–d). During mitosis, endogenous TPX2-1 was enriched in the spindle poles and at the edges of the phragmoplast, suggesting preferential binding to MT minus ends. The TPX2-1 signals gradually diminished during cytokinesis (Fig. 2a, b). TPX2-2 and TPX2-4 decorated the whole spindle, excluding the midzone (Fig. 2c, d). During anaphase, the midzone that was devoid of TPX2-2 and TPX2-4 expanded, suggesting a low affinity for anti-parallel MTs that bridge two halves of the spindle/phragmoplast. Both TPX2-2 and TPX2-4 localized to the early phragmoplast devoid of the midzone, but their signals decreased as cytokinesis progressed (Fig. 2c, d and Supplementary Movie 1). Similar localization has been reported for *Arabidopsis* TPX2[16]. TPX2-5, similar to TPX2-2 and TPX2-4, was detected in the spindle except in the midzone; however, the signal was much weaker, particularly in the polar region, and could be seen only when contrast/brightness was adjusted (Fig. 2e, max intensity panel). Unlike other TPX2s, TPX2-5 was strongly upregulated in anaphase and decorated the phragmoplast (except for the midzone) throughout cytokinesis (Fig. 2e and Supplementary Movie 1).

**Generation of hypomorphic mutants of TPX2**. The similarity in amino acid sequences suggested that TPX2-1 to −4 have redundant functions. Therefore, we simultaneously targeted these genes using a previously established CRISPR/Cas9 protocol[23]. We isolated a line, named *TPX2 1-4Δ*, in which frameshifts were introduced to all four *TPX2* genes in the exons present in all transcript variants identified in the Phytozome database (Supplementary Fig. 1a). The *TPX2 1-4Δ* line developed protonema and gametophores in a similar manner to the parental "GH" line (Fig. 3a and Supplementary Fig. 2). We then attempted to knock out the *TPX2-5* gene in the *TPX2 1-4Δ* background by means of homologous recombination. However, we could not isolate a knockout line after multiple attempts.

Nonetheless, in our attempt to knockout the *TPX2-5* gene, we isolated three lines with defective gametophore development after two independent transformations (Fig. 3a and Supplementary Fig. 2). In these lines, the original *TPX2-5* gene was replaced with a hygromycin cassette, as confirmed by PCR and sequencing (Supplementary Fig. 1b, c). However, DNA was amplified by PCR using *TPX2-5* "internal" primers, and we confirmed by sequencing that all exons of the *TPX2-5* gene remained (Supplementary Fig. 1d). These data suggested that the *TPX2-5* gene removed from the original locus was re-inserted into another genomic locus, possibly through micro-homology recombination. In all three mutants we isolated, the expression of *TPX2-5* mRNA was strongly reduced (Supplementary Fig. 1e). Hereafter, these hypomorphic lines are referred to as "*TPX2-5 HM1*," "*TPX2-5 HM2*" and "*TPX2-5 HM3*".

To test possible redundancy between *TPX2-5* and other *TPX2*s, we performed a rescue experiment in which full-length *TPX2-4* (with Cerulean tag) was expressed at the native locus in the *TPX2-5 HM1* line (Supplementary Fig. 3). These lines no longer showed a mitotic delay or dwarf gametophore phenotype (Supplementary Figs. 2b and 3b–d). In addition, we attempted to knockout the *TPX2-5* gene in the wild-type background (i.e.,

other *TPX2* genes were intact). Once again, we isolated a *TPX2-5* translocation line termed *TPX2-5 M* (Supplementary Fig. 2). However, in the wild-type background, relocation of the *TPX2-5* did not yield any noticeable phenotypes, which is consistent with the result of the rescue experiment (Supplementary Fig. 2). Next, we tried to isolate a frameshift or a mutant with a large gene deletion in the wild-type background using CRISPR. gRNAs that simultaneously targeted the first and fifth exons of *TPX2-5* were transformed (Supplementary Fig. 4a). We isolated two colonies, named *TPX2-5 Cr#30* and *TPX2-5 Cr#33*, that had a large portion of *TPX2-5* gene eliminated (Supplementary Fig. 4b). However, these lines also did not have any observable defects during cell division in protonema ($n = 15$) or gametophores ($n = 21$) (Supplementary Fig. 4c–e). These results suggest that five *TPX2* genes in *P. patens* are, at least partially, redundant for moss development during the protonema and gametophore stages.

**Spindle motility and division plane shift in the *TPX2* mutant gametophore**. Next, we aimed to characterize the dwarf gametophore phenotype in more detail (Fig. 3a). The *TPX2-5 HM1* line had smaller gametophore phyllids (hereafter called leaves), fewer cells per leaf, and reduced cell density (Fig. 3b–e), suggesting that cell proliferation decreased during leaf development. In 14 out of 23 leaves, the specialized central conducting cells of the midrib or more narrow cells on the edge of the leaf could not be visually distinguished (Fig. 3b). The cell size was analyzed separately in the apical and basal sides of the gametophore leaves (Fig. 3f). The apical side of the leaf contains dividing cells of a smaller size, which later undergo expansion and become larger cells at the basal side[24]. We observed that *TPX2-5 HM1* leaves had abnormally larger cells on the apical side, but not on the basal side of the gametophore, compared to controls (Fig. 3f). Occasionally, we observed extremely large cells in the leaves of the *TPX2-5 HM1* line. These cells were 3–4 times larger than the surrounding cells (Fig. 3b, arrowhead). The increase in cell size on the basal side of the leaf appeared to correlate with higher ploidy levels in GH and *TPX2 1-4Δ* lines, as revealed by DNA staining with the fluorescent dye DAPI (Fig. 3f, g). However, the difference in cellular DNA content between the apical and basal sides was less pronounced in the *TPX2-5 HM1* line. These data suggest that *TPX2-5 HM1* has fewer cell divisions during gametophore development and may have defects in ploidy control.

Next, we investigated whether TPX2 plays a role in mitosis of gametophore cells. Dwarf organ development in plants is sometimes associated with defective cytokinesis[25]. Thus, we used the lipophilic dye FM4-64 to visualize cell plates in gametophore initials (stem cells) after the first cell division. We observed that the position of the cell plate shifted to the basal side of the gametophore initial in multiple cells in all three *TPX2-5* mutants, dramatically skewing the size ratio between apical and basal daughter cells (Fig. 4 and Supplementary Movie 2). This was due to an overall reduced level of TPX2, as the cell plate positioning was normal in the aforementioned *TPX2-4* rescue lines and *TPX2-5 M* line (Fig. 4 and Supplementary Movie 2). To investigate what caused defects in the cell division site in the *TPX2-5 HM1,2,3* lines, we next performed live-cell imaging of the first cell division in the gametophore initial.

To observe mitotic MTs in gametophore initial cells, we introduced another marker, mCherry-tubulin, to *TPX2-5 HM1*, *TPX2 1-4Δ*, and control GH lines (note that histone and tubulin were labeled with the same color). The majority of gametophore initial cells in the *TPX2-5 HM1* line formed a bipolar spindle (90%; $n = 19$; Fig. 5a, b), although slightly smaller than the spindle in the control or *TPX2 1-4Δ* lines (Fig. 5c). Strikingly, we observed that the bipolar metaphase spindle moved

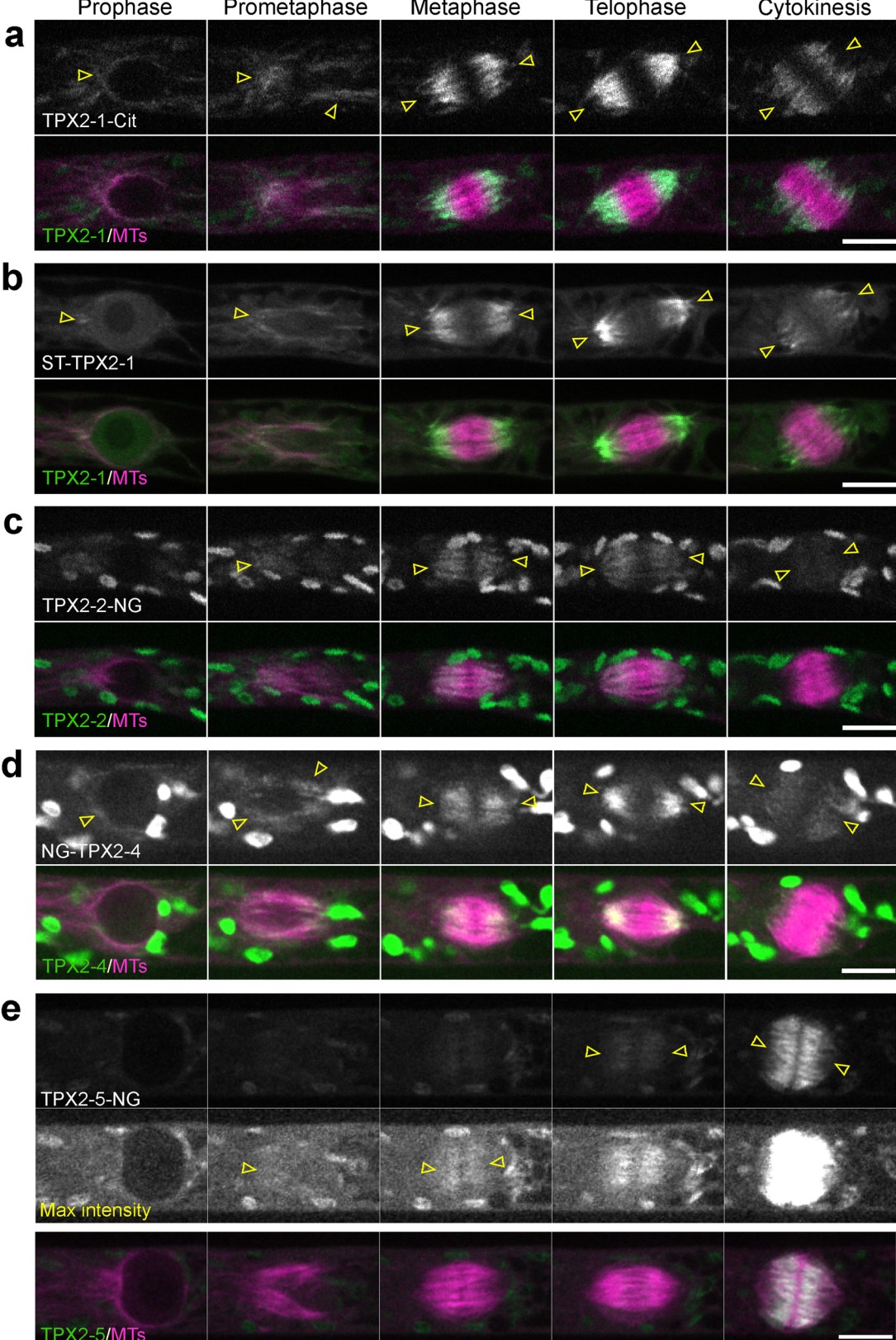

**Fig. 2 Localization of TPX2 proteins during mitosis.** Live-cell imaging was performed in caulonemal apical cells of *P. patens*, expressing mCherry-tubulin and one of the following TPX2 proteins endogenously tagged with a fluorophore: **a** TPX2-1-Citrine; **b** SunTag-TPX2-1; **c** TPX2-2-mNeonGreen; **d** mNeonGreen-TPX2-4; **e** TPX2-5-mNeonGreen. The SunTag-TPX2-1 line also expressed scFv-GCN-sfGFP under a β-estradiol-inducible promoter. Localization was observed in 2 independent experiments with similar results. Arrowheads indicate the fluorescent tag-based signals, whereas other ellipsoidal signals outside the spindle/nucleus represent autofluorescent chloroplasts. Bars, 10 μm.

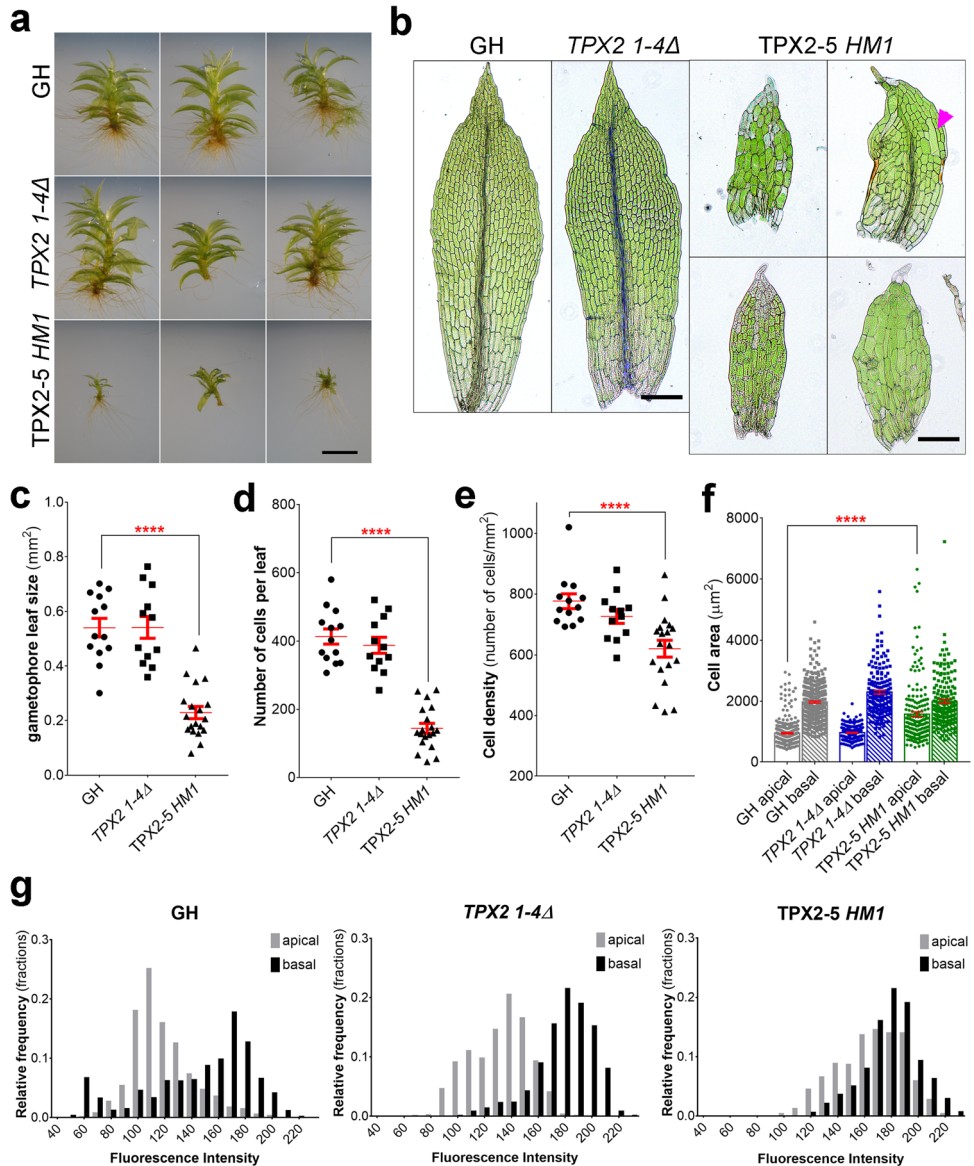

**Fig. 3 Abnormal gametophore and leaf development in the *TPX2-5 HM1* mutant. a** Representative images of gametophores after 4 weeks of culture of GH (control), *TPX2 1-4Δ*, and *TPX2-5 HM1* lines. Brightness/contrast have been linearly adjusted. An extended version is presented in Supplementary Fig. 3. Experiment was repeated three times with similar results. Bar, 2 mm. **b** Representative images of gametophore leaves of GH, *TPX2 1-4Δ*, and *TPX2-5 HM1* lines. The magenta arrowhead indicates abnormally large cell occasionally observed in the *TPX2-5 HM1* line. Bar, 200 μm. **c** Gametophore leaf size (mm²), **d** number of cells per leaf, **e** cell density (number of cells per mm²), **f** cell area (μm²) of the apical and basal sides of the gametophore, measured in gametophore leaves collected after 3 weeks of culture with $n = 13$, 12, and 19 for GH, *TPX2 1-4Δ*, and *TPX2-5 HM1* lines, respectively (mean ± SEM, ****$p = 0.0001$ by one way ANOVA with Dunnett's multiple comparison test against GH). Apical and basal sides correspond to 1/3 of leaf length at the tip and at the base, respectively. **g** Quantification of the nuclear DNA content in the interphase nucleus of gametophore leaf cells (from apical and basal sides). DNA amounts were measured as fluorescence intensity (arbitrary unit) of the DAPI-stained nuclei per cell without background subtraction. GH and *TPX2 1-4Δ* lines have two ploidy peaks corresponding to apical and basal sides of gametophore leaves; In the *TPX2-5 HM1* line, the difference in ploidy is less pronounced.

unidirectionally over 5 μm to the basal side in 73% ($n = 14$) of *TPX2-5 HM1* cells (Fig. 5a, d and Supplementary Movie 3). Consequently, the phragmoplast formed close to the basal edge. We also monitored the two- and four-cell stages of the gametophore, in which the first division plane was observed at the normal position. We observed spindle motility in six of ten cell divisions (Supplementary Movie 4). We concluded that defective spindle positioning after NEBD causes abnormal cell plate position during gametophore formation in the *TPX2-5 HM1* line.

**Spindle motility in the *TPX2* mutant is actin-dependent**. In animals, spindle motility relies on MTs and actin. To test the involvement of the cytoskeleton in spindle motility, we first partially depolymerized MTs in the *TPX2-5 HM1* line using low-dosage oryzalin (200 nM), a MT-destabilizing drug. Upon treatment, the spindle behaviors were more variable and showed overall differences from the untreated cells. Most notably, in 5 of 16 cells, we observed that the spindle had shifted towards the apical side of the cells, which was not observed in untreated cells (Fig. 5d). In addition, the maximum spindle speed was decreased

# a

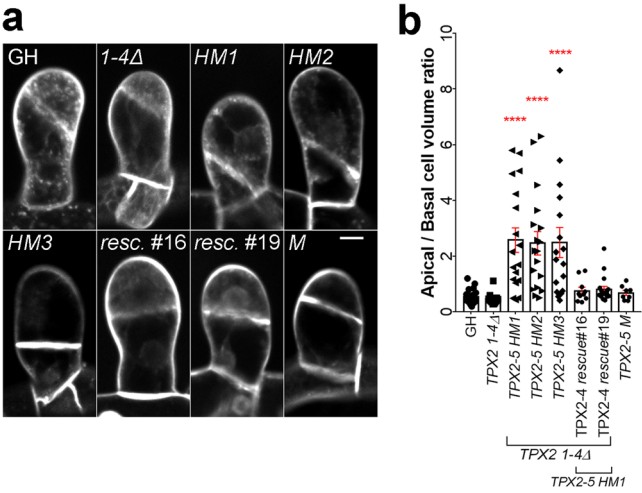

**Fig. 4 Abnormal cell plate positioning in the gametophore initials of the TPX2-5 hypomorphic mutants. a** Gametophore initial at the 2-cell stage stained with FM4-64 dye. Bar, 10 μm. **b** The apical/basal cell volume ratio was estimated as the apical cell volume divided by the basal cell volume, measured during the 2-cell stage (mean ± SEM, ****$p = 0.0001$ by one way ANOVA with Dunnett's multiple comparison test against GH). $n = 22$, 15, 18, 18, 17, 10, 18, and 8 for GH, *TPX2 1-4Δ*, *TPX2-5 HM1*, *TPX2-5 HM2*, *TPX2-5 HM3*, TPX2-4 *rescue*#16, TPX2-4 *rescue*#19 and *TPX2-5 M*.

(Fig. 5e). In the presence of oryzalin, 9 out of 11 spindles moved slower than 1 μm/min (maximum speed), whereas only 2 out of 13 did so in control cells (DMSO). These data suggest that MTs contribute to spindle motility to a certain extent.

We also tested the effect on spindle motility of the MT-stabilizing drug taxol at a final concentration of 10 μM. The frequency of cells showing spindle motility did not decrease (18 of 23 cells). However, the motility was slower and the traveling distance was shorter than that of the control cells (Fig. 5e–g). The difference was not statistically significant, possibly because the spindle motility phenotype showed high variability in the *TPX2-5 HM1* line. This positive but mild effect might be due to the low sensitivity of *P. patens* to taxol[26]. For example, we never observed mitotic arrest after taxol treatment, which is expected after high-dose taxol treatment of animal cells[27].

Next, we investigated the effect of actin depolymerization with latrunculin A on the spindle motility of *TPX2-5 HM1* cells. Strikingly, the high-dose latrunculin A treatment (5 μM) completely suppressed the basal motility of the spindle in the *TPX2-5 HM1* line (20 of 20 cells, Fig. 5a, d, Supplementary Movie 3). Low-dosage treatment (50 nM) was also effective; only 2 out of 18 cells exhibited spindle translocation. Notably, actin depolymerization by latrunculin did not affect spindle morphology, orientation, or positioning in wild-type cells[13]. Thus, actin is the driving force of basal spindle motility in the *TPX2* mutant.

Next, we attempted to visualize actin distribution during spindle movement. We introduced Citrine-F-tractin (actin marker) in the wild-type and *TPX2-5 HM1* background, expressing mCherry-tubulin or mCherry-tubulin/Histone-RFP, respectively (Fig. 6 and Supplementary Movie 5). Cortical actin was observed as cables in both lines and was distributed evenly around the gametophore initial cell (Fig. 6, Z-projection). We did not detect any difference in the cortical actin networks between the control and *TPX2-5 HM1* lines. Citrine-F-tractin signals were broadly distributed in the cytoplasm. Unlike the cortical actin network, we only occasionally observed prominent actin cables and bright speckles in the cytoplasm, suggesting that the cytoplasmic actin network comprised less structured filaments (Fig. 6a, indicated by arrowheads). It is also possible that some of

the cytoplasmic signals originated from Citrine-F-tractin not bound to the actin filaments. Cytoplasmic actin remained after NEBD in both the control and *TPX2-5 HM1* lines. Some areas, mostly on the basal side of the cells, were devoid of Citrine-F-tractin signals; they were most likely occupied by vacuoles. In the *TPX2-5 HM1* mutant undergoing spindle motility, cytoplasmic actin signals moved with the spindle (Fig. 6a and Supplementary Movie 5). Since cytoplasmic streaming is absent in *P. patens*[28], we interpret that the apparent movement of the cytoplasm is passive, following spindle motility. When the Citrine-F-tractin signal intensity was compared between the apical and basal poles of the spindle, we did not observe a significant difference in either the wild-type or *TPX2* mutant (Fig. 6b).

We also tested whether the overall gametophore development could be restored by mild actin disruption. *TPX2-5* mutant lines were grown for 4 weeks on agar plates and a range of latrunculin A was supplied (50 pM–100 nM). Although protonema growth was inhibited at higher concentrations of latrunculin A, the gametophore phenotype was not rescued under any condition (Supplementary Fig. 5).

**Roles of TPX2-5 in spindle assembly in protonema**. The Physcomitrella protonema consists of tip-growing chloronemal and caulonemal cells. ACD occurs also in this tissue. The apical, tip-growing cells frequently divide, while subapical cells occasionally form a bulge and divide asymmetrically, producing a new apical tip cell, in a process known as "branching"[29]. We did not observe spindle motility in the protonema of the *TPX2-5 HM1* line ($n = 85$, including caulonemal/chloronemal apical cells and branching cells). The identified differences from the wild-type were the cell plate orientation and apical cell growth speed in caulonemal cells, which might reflect a slight alteration in MT dynamics in the mutant (Supplementary Fig. 6). However, it is unlikely that these protonema defects have a significant impact on gametophore development.

A rare spindle phenotype observed in both protonema ($n = 1$ of 85) and gametophore initial cells ($n = 2$ of 19) was spindle collapse, which led to chromosome missegregation, cytokinesis failure, and multinucleation (Supplementary Movie 6). In an earlier study, we showed that a certain percentage of protonema cells of *P. patens* were able to recover after cytokinesis failure and re-enter the cell cycle, resulting in whole genome duplication[30]. In the context of gametophore development, diploid *P. patens* have fewer gametophores for unknown reasons[31]. To exclude the possibility that *HM* mutants have a high number of diploid protonema cells, compromising gametophore development, we conducted an analysis of ploidy levels by measuring fluorescence intensity of DAPI-stained nuclei in the protonema cells of control (GH), *TPX2 1-4Δ*, and *TPX2-5 HM1,2,3* lines. We detected no additional peaks that would indicate higher ploidy in any of these lines (Supplementary Fig. 7).

To gain insights into the essential mitotic function of TPX2, we performed a detailed analysis of cell division phenotypes in the mutant with the greatest effect. To this end, we selected an inducible *TPX2-5* RNAi line in the *TPX2 1-4Δ* background. Since RNAi induction almost completely inhibited cell growth and gametophore development, we focused on the division of the protonemal apical stem cells that appear earlier than gametophores. Using time-lapse imaging of MTs and chromosomes, we observed severe MT phenotypes during the early stages of mitosis (Fig. 7a–d). During prophase, we detected a reduction in the signals of perinuclear MTs (Fig. 7e), which was also observed with a γ-tubulin RNAi[32], and abnormal nuclear shape, which is unique to the *TPX2 1-4Δ* mutant (elongated nucleus prior to NEBD, Fig. 7c). The nuclear elongation was not actin-dependent, as the

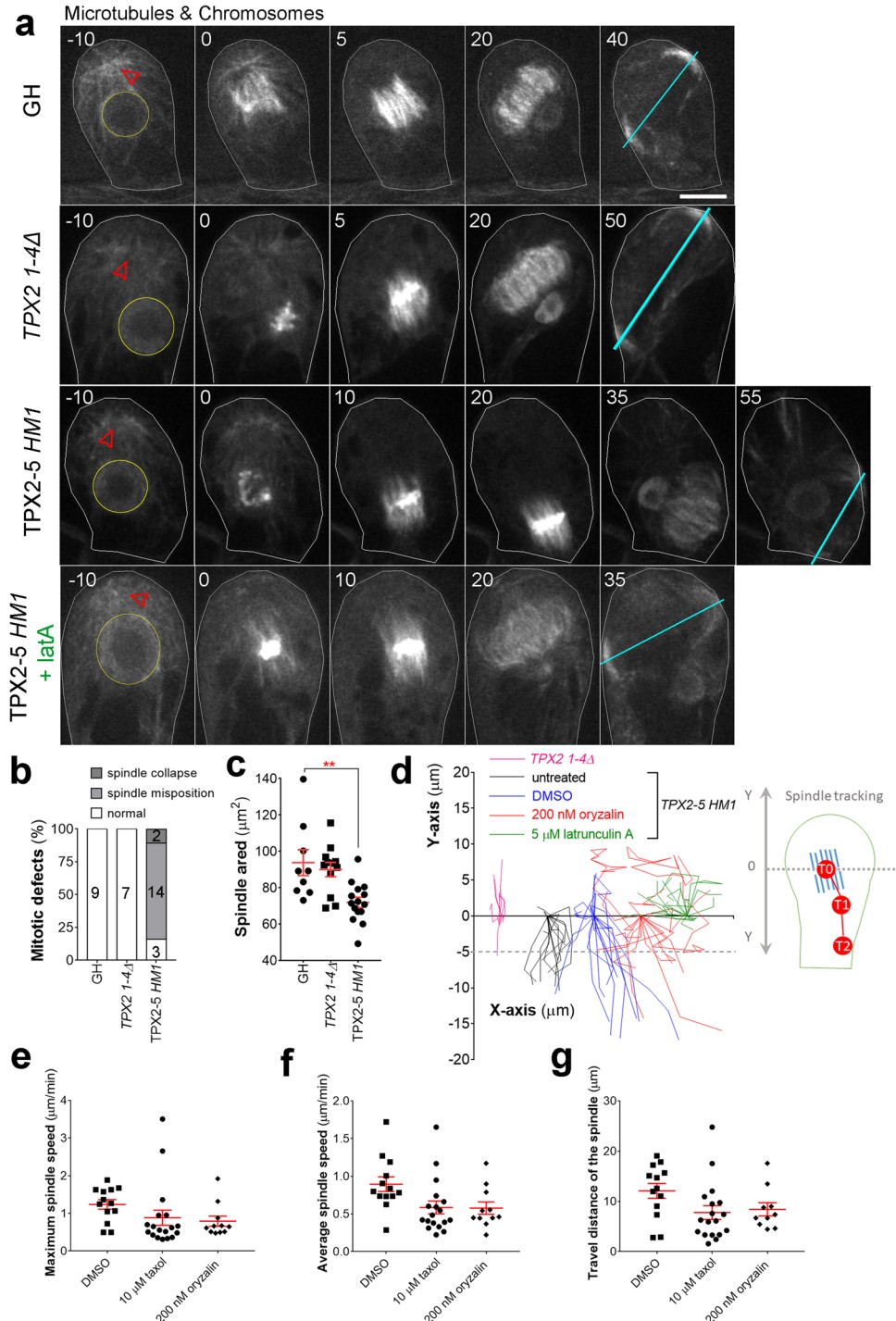

**Fig. 5 Spindle position is actively maintained through the interplay between microtubules and F-actin. a** Live-cell imaging of the first asymmetric division in the gametophore initial. The positions of the nucleus and gametosome (prophase MTOC appeared in the apical cytoplasm) are indicated with yellow circles and red arrowheads, respectively. Cyan lines show the position and orientation of the phragmoplast. Cell borders are outlined with white lines. Live-cell imagining was repeated three times with similar results. Bar, 10 μm. **b** The frequency and type of spindle defects in gametophore initial mitosis observed in GH (control), *TPX2 1-4Δ*, and *TPX2-5 HM1* lines. Numbers within the columns indicate number of cells with corresponding phenotypes. **c** Area occupied by the metaphase spindle (spindle size) in gametophore initials measured by manually tracking spindle borders from Z-projection, n = 9, 12, and 15 for GH, *TPX2 1-4Δ* and *TPX2-5 HM1* lines, respectively. (mean ± SEM; **p = 0.0029, two-tailed Student's t-test). **d** Tracking of the spindle center position from NEBD to anaphase onset. We assigned the starting position as Y = 0 and different X positions for each sample group. Note, that after 5 μM latrunculin A treatment, spindles never showed motility towards the basal end of the cell, i.e., negative Y-values. Each line represents spindle movement in a single cell. More than 12 cells were observed for each sample group in three or more independent experiments. **e** Maximum spindle speed (μm/min), **f** average spindle speed (μm/min), and **g** distance traveled by the spindle (μm) in the *TPX2-5 HM1* cell line under various treatments. Total number of cells observed, including those without spindle motility: n = 23 (10 μM taxol), 16 (200 nM oryzalin), and 20 (control DMSO, with which the stock solution of taxol, oryzalin, and latrunculin A were prepared). Only spindle motility towards the basal end of the cell was analyzed. Bars represent mean ± SEM.

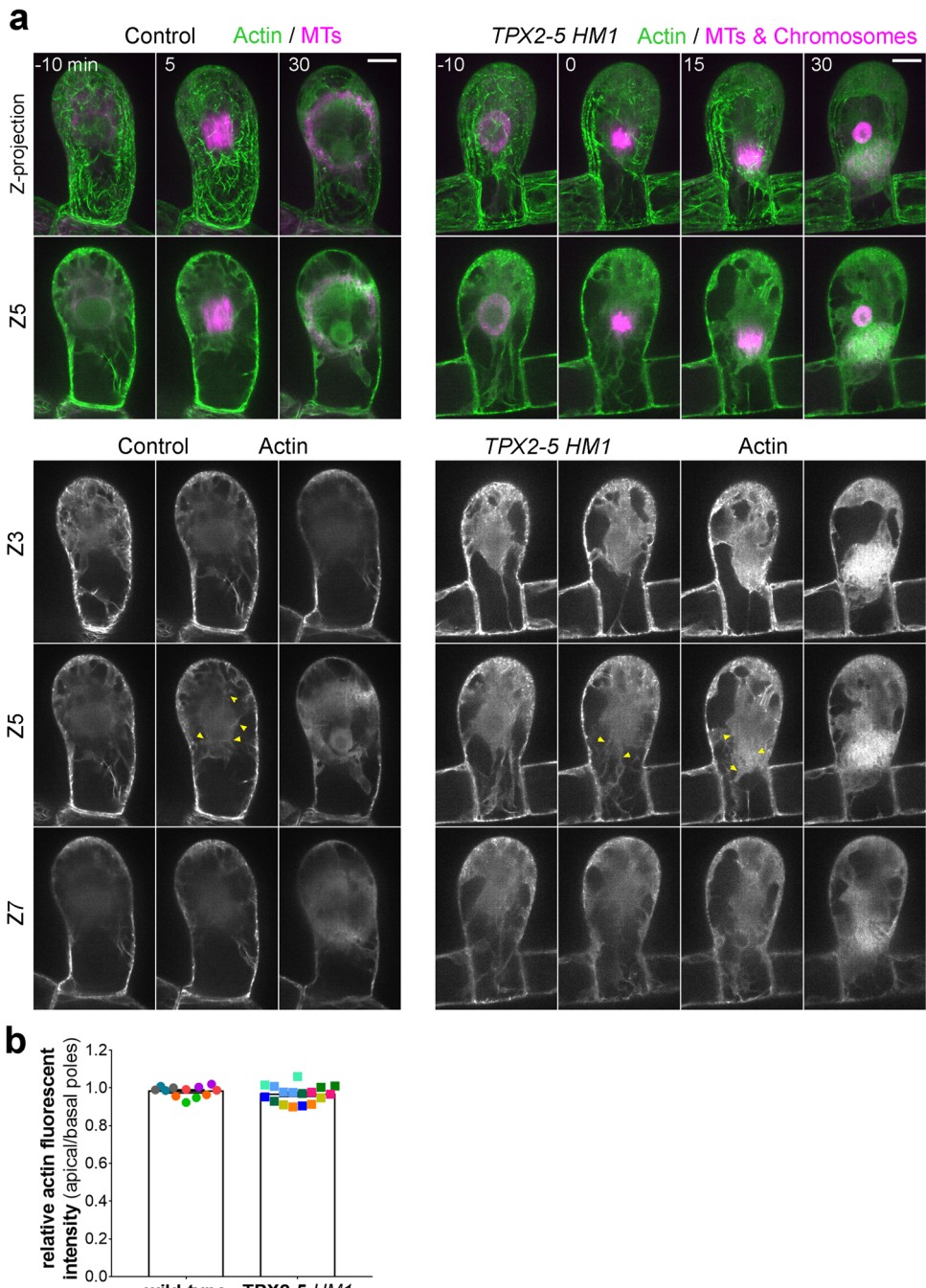

**Fig. 6 Actin is present in the cytoplasm and cortex during spindle motility in the *TPX2-5 HM1* line. a** Live-cell imaging of Citrine-F-tractin (actin, green) during the first asymmetric division in the gametophore initial cells also expressing mCherry-tubulin (magenta, MTs). *TPX2-5 HM1* cells additionally expressed mCherry-tubulin/H2B-RFP (magenta, chromosomes), which was brighter than mCherry-tubulin. Maximum intensity projection of Citrine-F-tractin (20 μm with 2.5 μm steps) and single focal frame of mCherry-tubulin were merged in the "Z-projection" panels. Three Z frames are also presented (z = 3, 5, and 7). Yellow arrowheads indicate filamentous or punctate signals of cytoplasmic actin. Time 0 (min) was set at NEBD. Bar, 10 μm. **b** Relative actin fluorescent intensity (basal/apical poles). Mean gray values of Citrine-f-tractin were measured at the apical and basal pole of the spindle in a square of a fixed size (8 × 4 μm) for each of the Z-slices and summed to get intensity at each pole. Dots of the same color represent measurements obtained from the same cell at different time points (n = 6 and 8 for wild-type and *TPX2-5 HM1*, respectively).

length in prophase was not significantly changed by latruculin A treatment (17.8 ± 2.8 and 17.6 ± 2.5 μm before and after treatment, respectively [5 μM, 10 min]). After NEBD, 3 out of 52 cells in the RNAi lines failed to form bipolar spindles, followed by metaphase arrest and chromosome missegregation, which was never observed in the control lines (Fig. 7d). Other cells formed bipolar spindles; however, the MT signals in the prometaphase spindle was greatly reduced (Fig. 7f, blue). This was no longer the case at metaphase, where the intensity of MT signals was similar to control spindles, indicating the recovery of MT numbers during prometaphase (Fig. 7f, green). Other phenotypes observed upon TPX2-5 depletion include chromosome missegregation (29%, n = 15) and spindle misposition/orientation (33%, n = 17, Fig. 7d and Supplementary Movie 7).

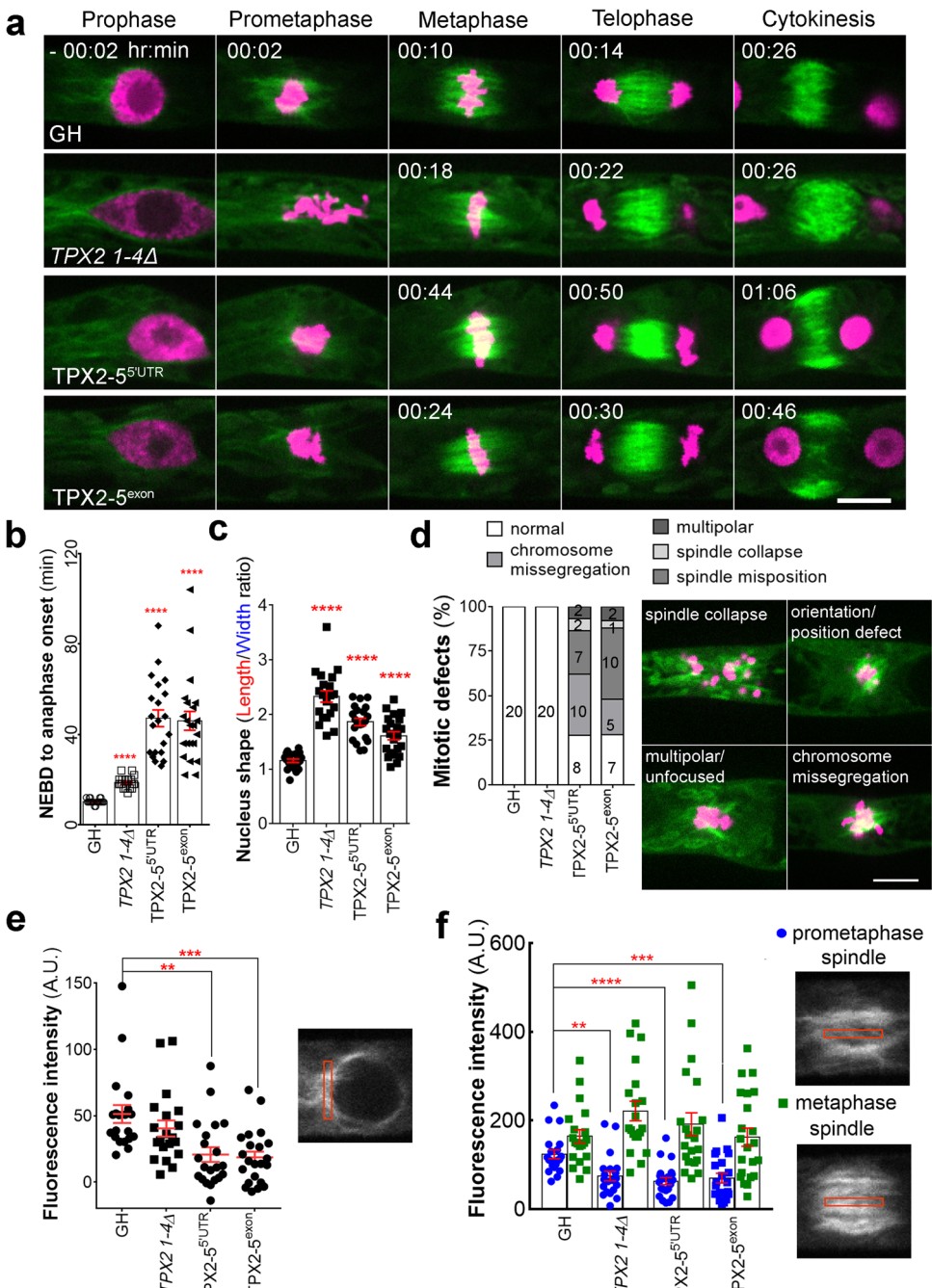

**Fig. 7 TPX2 contributes to microtubule amplification in early mitosis. a** Representative images of the mitosis of protonemal apical cells in GH (control), *TPX2 1-4Δ*, *TPX2-5⁵ᵁᵀᴿ* RNAi, and *TPX2-5ᵉˣᵒⁿ* RNAi lines. Time 0 (min) was set at NEBD. Green, GFP-tubulin; Magenta, histoneH2B-mRFP. Bar, 10 μm. **b** Mitotic duration calculated from NEBD to anaphase onset (mean ± SEM; ****$p < 0.0001$, two-tailed Student's t-test). Live-cell imagining was repeated three times with similar results. **c** Nucleus shape prior to NEBD, measured as a ratio of nucleus length to nucleus width (mean ± SEM; ****$p < 0.0001$, two-tailed Student's t-test). **d** Frequency and type of mitotic defects observed. Numbers within the columns indicate number of cells with corresponding phenotypes. Bar, 10 μm. **e** Fluorescence intensity of perinuclear MTs (mean ± SEM; **$p = 0.0011$, ***$p = 0.0002$; two-tailed Student's t-test). A.U. stands for Arbitrary Units. **f** Fluorescence intensity of MTs in the prometaphase spindle (4 min after NEBD) and metaphase spindle (2 min before anaphase onset), measured from a single focal plane, with the cytoplasmic background subtracted. A decrease in fluorescence intensity was detected in prometaphase, but not in metaphase spindles (mean ± SEM **$p = 0.0018$, ****$p = 0.00002$, ***$p = 0.0008$; two-tailed Student's t-test). For graphs **b–f**, number of independent cells observed $n = 20$ for GH and *TPX2 1-4Δ* lines, $n = 21$ for TPX2-5⁵ᵁᵀᴿ, and $n = 22$ for TPX2-5ᵉˣᵒⁿ RNAi lines.

## Discussion

This study of the moss, Physcomitrella, aimed to functionally characterize TPX2, one of the most well-known spindle MT-associated proteins in animals, taking advantage of the available genetic and cell biological tools[33] to comprehensively examine highly duplicated mitotic genes in plants. We identified defects in spindle morphology and chromosome segregation associated with *TPX2* depletion and observed unexpected spindle motility during ACD in the *TPX2-5* mutant.

Our functional characterization suggests that, despite certain differences in localization, the five TPX2 proteins of *P. patens* are partially redundant. Frameshift mutations could be introduced

into individual *TPX2* genes. Furthermore, the expression of TPX2-4 protein fully rescued the phenotype of the *TPX2-5 HM1* mutant, in which *TPX2 1-4* genes had frameshift mutations and *TPX2-5* expression was reduced. Our results in protonemal cells suggest the functional conservation of moss TPX2 with well-studied animal orthologs, namely assisting in MT formation through nucleation and/or stabilization during spindle assembly. Since the γ-tubulin activation motif [21] is partially conserved in all moss TPX2 proteins (Fig. 1b), TPX2 may be required for γ-tubulin-dependent MT nucleation. For the cells that assembled bipolar spindles after TPX2 depletion, phragmoplast formation and expansion were similar to those in control cells. During cytokinesis in protonemal cells, branching MT nucleation takes place, which depends on γ-tubulin and augmin complexes[32]. We speculate that this mechanism nucleates a sufficient number of MTs, even when TPX2 is reduced. However, our results do not yet suggest a clear role for TPX2 in post-anaphase. Severe spindle phenotypes, such as spindle collapse or multipolar formation, were observed upon *TPX2-5* RNAi, and the post-anaphase function could not be assessed. Whether and which *TPX2* genes are critical in later developmental stages also remains to be determined.

Gametophores were small in *TPX2-5* mutants and the leaf cells showed multiple abnormalities in morphology and ploidy. Defects in cell expansion could be a possible explanation for the dwarfism, as gametophore cells grow by expansion, controlled by turgor pressure and cytoskeleton dynamics. However, it does not appear to be the cause of smaller gametophores in the *TPX2* mutant, since leaf cells were not overall smaller than controls. Although we cannot rule out the possibility that non-mitotic functions of TPX2 might be required for proper gametophore development, our data are consistent with the notion that defective cell division underlies this phenotype. In the mutant there were fewer cells per leaf, suggesting that fewer cell divisions occurred during gametophore development. Cell division defects were not exclusive to the first division of the gametophore initial; they were sometimes observed in the second and subsequent divisions (Supplemental Movie 4). Therefore, it is plausible that division plane mispositioning occasionally occurs later during gametophore development. We propose that spindle motility and, as a consequence, skewed daughter cell ratio and/or misplaced cell plates have a long-reaching and cumulative effect on gametophore development. For example, an extremely asymmetric division due to spindle translocation would give rise to a giant cell with abnormal DNA/cytoplasm ratio, which might affect its cell cycle or DNA replication control, and an overly small cell, that may not be able to divide again. Additionally, we observed spindle collapse followed by cytokinesis failure in ~10% of gametophore cells. In the context of organ development, these cells would have a higher ploidy and might overexpress certain genes[34], affecting surrounding cells, as well.

In plants, defective division sites have been mostly attributed to defects in PPB formation[11,35], phragmoplast guidance errors (CDZ deficiency)[10,36], or abnormal positioning of the nucleus in prophase[37,38]. The current study identified an independent and hitherto unappreciated cause of division site abnormality: spindle motility after NEBD. Spindle-specific positioning defects in meiosis II have been observed in *jas* and *ps1* mutants of Arabidopsis due to abnormal organelle distribution[39]. However, meiosis II is a unique system in which two spindles share a cytoplasm and may be partially fused in the absence of an organelle barrier. Therefore, there is unlikely to be a mechanistic analogy to moss gametophores. The spindle motility was observed in the gametophore initial in Physcomitrella *TPX2* mutants. A recent study using neural stem cells of the embryonic developing mouse neocortex showed that TPX2 knockdown not only affects spindle MT generation but also spindle orientation, implicating a conserved TPX2-dependent mechanism of spindle positioning[40]. However, the mechanism by which TPX2 functions in spindle positioning remains unclear at a mechanistic level. We also observed unstable spindle position in protonemal cells upon TPX2 depletion with RNAi (Fig. 7d and Supplementary Movie 7). In RNAi lines, the spindle positioning defect was almost always accompanied by chromosome missegregation or other phenotypes, complicating the interpretation of the results. Nonetheless, active spindle positioning may not be unique to gametophores.

In addition to TPX2, we here uncover the involvement of actin in spindle positioning. This was also an unexpected observation, as the function of actin in plant cell division has been mostly attributed to phragmoplast guidance during cytokinesis[4,41]. Actin is known to play an important role in spindle positioning in animal cells[42]. Of particular interest are animal oocytes, as they lack centrosomes, like plant cells do. In mouse oocytes, spindle migration and symmetry breaking are driven by changes in the stability of the actin meshwork, the formation of which depends on actin nucleators, such as formin-2, and myosin II motor[42,43]. We observed cytoplasmic and cortical actin in the gametophore initial cell. Thus, an analogous mechanism might transmit actin-dependent force to transport spindles in oocytes and moss. However, it should be noted that the actin role emerged only in the background of a reduced *TPX2* function. We suggest that there is "tug-of-war" between spindle and actin-dependent forces (Fig. 8). In wild-type moss cells, sufficient numbers of spindle MTs may predominate in the tug-of-war against actin-dependent forces. Conversely, when the overall levels of TPX2 decreased, fewer and/or less stable MTs may be insufficient to hold the spindle in place. This model is supported by the observation that a less severe spindle motility phenotype was observed in the mutant after taxol treatment, which partially stabilized MTs[44]. In contrast, oryzalin might further attenuate the interaction between actin and MTs, resulting in misoriented and slow spindle movement. The exact mechanism that transmits the force from the actin network to the acentrosomal spindle and whether this phenomenon is conserved in other plant species remains an open question for future investigation.

## Methods

***P. patens* culture and transformation**. *P. patens* culture and transformation protocols have been described in detail in[17]. In brief, BCDAT agar medium (BCDAT stands for stock solutions B, C, D, and ammonium tartrate) was used for regular *P. patens* culture at 25 °C under continuous light. Transformation was performed using protoplasts. To prepare protoplasts, the protonema tissue, collected after 7 days of culture, was incubated in the solution containing 2% driselase and 8% (w/v) mannitol for 30 min at room temperature with gentle rocking. The solution was filtered using 50 μm nylon-mesh and centrifuged at $180 \times g$ for 2 min to collect protoplasts. The supernatant was discarded, and protoplasts were washed twice in 20 ml 8% mannitol. Protoplast concentration before transformation was adjusted to $1.6 \times 10^6$ ml$^{-1}$ by adding MMM solution (0.1 % MES pH 5.6, 15 mM MgCl$_2$, 9.1% (w/v) mannitol). Next, 300 μL of protoplast solution were mixed with 30 μL of plasmid (30 μg) and 300 μL of 40% (w/v) polyethylene glycol solution (10 mM Tris-HCl pH 8.0, 100 mM Ca(NO$_3$)$_2$, 8% mannitol) and incubated first for 5 min at 45 °C, then for 10 min at 20 °C, followed by incubation in the dark at 25 °C overnight. Protoplasts were then spread onto PRM agar plate on which sterile cellophane was placed. After three days of culture, the cellophane with regenerated protoplasts was transferred to the BCDAT medium that contained appropriate antibiotics for selection. After another round of passage onto drug-free and drug-containing BCD media, the target integration was confirmed by genotyping PCR and/or sequencing in the case of CRISPR-generated lines. The plasmid containing drug-resistant cassette was eliminated from the cell during CIRSPR line selection. The GH line, expressing histone H2B-mRFP and GFP-α-tubulin, was used for transformation in the CRISPR and knockout experiments, while the mCherry-α-tubulin #52 line was used for Citrine, mNeonGreen endogenous tagging, and for Citrine-F-tractin transformation as a control. The transgenic lines generated in this study are listed in the Supplementary Data 1.

**Molecular cloning**. Plasmids for Citrine or mNeon-Green endogenous tagging were assembled using In-Fusion (Clontech, Mountain View, CA, USA), in which

**Fig. 8 Schematic representation of the spindle impact on phragmoplast position and orientation in the gametophore initial cells of _P. patens_.** The upper panel shows the first asymmetric division of gametophore initial in wild-type cells. Spindle, consisting of microtubules, drives chromosome segregation and cell plate expansion. Cytoplasmic actin surrounds the nucleus before nuclear envelope breakdown and the spindle in early mitosis. During cytokinesis, cytoplasmic actin localizes to the phragmoplast. The bottom panel summarizes the findings of the present study and the role of the microtubule-associated protein TPX2 in spindle positioning. In the scenario where TPX2 function is reduced, the spindle can be transported to the bottom of the gametophore initial cell, compromising the cell plate position and daughter cell ratios in asymmetric cell division. The cytoplasm changes its position together with the moving spindle, and spindle motility is completely inhibited by depolymerizing actin filaments with latrunculin A (Lat A). Stabilizing microtubules with taxol could partially counteract spindle transport, suggesting that under normal conditions, microtubules are able to fix the spindle position against actin force.

Citrine or mNeonGreen genes, a G418 resistance cassette (only C-terminal tagging), and homologous recombination regions (500–800 bp of the respective genes) were connected. A similar strategy was used to assemble the knockout plasmid for _TPX2-5_, wherein a hygromycin resistance cassette was flanked by the 5′ and 3′UTR regions of the _TPX2-5_ gene. A detailed protocol for endogenous gene tagging and knockouts in _P. patens_ has been previously published[17]. The CRISPR protocol has been described in detail in[23]. In brief, CRISPR gRNAs targeting one of the exons were designed using the online tool, CRISPOR (http://crispor.tefor.net/), based on target gene specificity (off-target score) and predicted frameshift efficiency. Individual gRNAs were ligated into pCasGuide/pUC18 vector pre-digested with BsaI. Next, individual gRNAs together with the _U6_ promoter and gRNA scaffold region were amplified by PCR and assembled into a single multi-gRNA plasmid, also containing a hygromycin resistance cassette for transient plant selection. RNAi vectors were cloned using the Gateway system (Invitrogen, Carlsbad, CA, USA), with pGG624 as the destination vector. Two independent, non-overlapping RNAi constructs were prepared for each gene. The full list of plasmids and primers used in this study are shown in Supplementary Data 2.

**Gene expression analysis with quantitative real-time PCR (qRT-PCR).** To determine the expression levels of _TPX2-5_ and _TPX2-4_ genes, total RNA was extracted from 7-day-old protonema tissue using the innuPREP Plant RNA kit (Analytik Jena GmbH, Berlin, Germany) according to the manufacturer's instructions. Total RNA was treated with DNase I (Thermo Fisher Scientific) at 37 °C for 1 h to remove genomic DNA contamination, and RNA quality was checked by gel electrophoresis. For cDNA synthesis, we used TaqMan Reverse Transcription Reagents (Thermo Fisher Scientific) with random hexamers, according to the manufacturer's protocols. Primers were designed using the Universal Probe Library (Roche, https://lifescience.roche.com/en_de/brands/universal-probe-library.html#assay-design-center) and selected according to the lowest amount of off-target hits identified in the _P. patens_ transcriptome in Phytozome (www.phytozome.net). The efficiency of each primer pair was confirmed prior to analysis with a qPCR run using a series of cDNA dilutions (1:2) and a control without cDNA. Melting curves were analyzed to exclude primer pairs with off-

target amplification. The primer pairs used for qRT-PCR are listed in Supplementary Data 2. Samples using a cDNA equivalent to 50 ng of total RNA from corresponding lines were prepared in triplicate or, alternatively, two sets of triplicates for TPX2-5 in the _TPX2-5 HM1, HM2,_ and _HM3_ lines, with the SensiMix Kit and SYBR Green (Bioline, Luckenwalde, Germany). Negative controls without template addition, as well as reverse transcription controls (reverse transcription reaction carried out without Multiscribe™ RT), were performed for each primer pair and each line. Amplification was performed in 40 cycles at a melting temperature of 60 °C. qRT-PCR data were analyzed using LightCycler R 480 software (Roche). The relative expression levels of the gene of interest (GOI), TPX2-5, and TPX2-4, were compared to those of the control (ctrl), internal housekeeping genes EF1α (Pp3c2_10310V3.1), and the ribosomal protein L21 (Pp3c13_2360V3.1), and was calculated as $2^{(-\Delta CT)}$, wherein $\Delta C_T = C_T(\text{GOI}) - C_T(\text{ctrl})$ and primer efficiency was assumed to be 2. $C_T$ was defined as the cycle number at which each sample reached an arbitrary threshold[45].

**Sample preparation for live-cell imaging.** The sample preparation method for live cell imaging has been described in detail in a previous study[17]. In brief, a glass-bottom dish coated with a thin layer of BCD agar medium was inoculated with moss protonema and cultured under continuous light at 25 °C prior to observation (BCD stands for stock solutions B, C, and D. Each recipe can be found in ref. [17]). For gametophore induction, 1 μM of the synthetic cytokinin, benzylaminopurine, or 1 μM 2iP (2-isopentenyladenine), diluted in 1 mL of distilled water, was added to the 6–7-day-old colony and incubated for 10 min. Next, the remaining liquid was aspirated with a pipette, the dish was sealed, and the sample was cultured as described above for 20–24 h prior to gametophore imaging. Latrunculin A, taxol, or oryzalin was diluted in distilled water to final concentrations. Prior to drug treatment, most of the agar pad from the glass-bottom dish was cut and removed to minimize dilution. For RNAi induction, 400 μL of 5 μM β-estradiol, diluted in distilled water, was added to the pre-cultured protonema 4 d prior to observation. Although β-estradiol was previously supplemented directly to the agar medium[46,47], we found that it almost entirely inhibited cell growth in _TPX2-5_ RNAi lines; hence, the protocol was modified.

**Microscopy and data analysis**. Sample preparation is described above. Localization, RNAi, and most of the gametophore images were acquired using a Nikon Ti microscope (60 × 1.30-NA lens; Nikon, Tokyo, Japan) equipped with a CSU-X1 spinning-disk confocal unit (Yokogawa, Tokyo, Japan) and an electron-multiplying charge-coupled device camera (ImagEM; Hamamatsu, Hamamatsu, Japan). The microscope was controlled using the NIS-Elements software (Nikon). Imaging after FM4-64 staining, 10 µM taxol treatment, and low-dosage latrunculin A treatment (50 nM, 5 nM, and 500 pM) and imaging of actin using Citrine-F-tractin probe were performed using a ZEISS inverted microscope (25 × 0.8-NA lens or 63 × 1.4-NA lens, Carl Zeiss Microscopy, Germany) equipped with a CSU-X1 spinning-disk confocal unit (Yokogawa, Tokyo, Japan) and CCD camera (Photometrics Prime sCMOS). All imaging was performed at 22–24 °C in the dark, except for the first division of the gametophore initial, since gametophore development requires light (3 min light/2 min dark cycle). Imaging was performed at least twice for localization and at least three times for phenotypic characterization. To obtain quantitative values, the data from multiple experiments were combined because of insufficient sample numbers in a single experiment.

For single-leaf imaging (Supplementary Fig. 3d), we dissected gametophores using syringe needles and scissors to isolate single leaves. Leaves were mounted in a drop of water between two coverslips, and images were acquired with a Nikon Ti microscope (10 × 0.30-NA lens) in bright-field mode. Gametophore and moss colony images were acquired after 4 weeks of culture using a stereoscopic microscope (SZ2-ILST, Olympus Corporation, Tokyo, Japan) equipped with a digital camera (Axiocam ICc1; Carl Zeiss Microscopy, Germany) or a Canon EOS 400 camera, respectively. Image data were analyzed using ImageJ software (National Institutes of Health, Bethesda, MD, USA). Prism software was used to plot the graphs and perform statistical analyses (GraphPad, San Diego, CA, USA).

**Cell volume analysis**. Cell volume was analyzed in gametophore initial cells stained with FM4-64 dye diluted in distilled water to a final concentration of 10 µM. The dye was added to the live-imaging dish before acquisition, without cutting the agar. Imaging was performed immediately after FM4-64 application. Images were acquired as a Z-stack (40 µm, 0.97 µm step) using a ZEISS inverted microscope (25 × 0.8-NA lens) equipped with a CSU-X1 spinning-disk confocal unit (Yokogawa, Tokyo, Japan) and a CCD camera (Photometrics Prime sCMOS). Cell boundary predictions and segmentation were performed using PlantSeg software[48]. Default conditions of PlantSeg were used with the following modifications: images were rescaled using in-build function (for our images voxel size was 0.26 µm on the X–Y dimensions and 0.97 µm on the Z dimension) and cell minimum size was set at 400,000 voxels. The accuracy of the cell predictions was confirmed by comparing PlantSeg output files to raw images for each cell. In some cases, when cell boundaries were not predicted accurately, images were deconvolved using Huygens Professional version 19.04 (Scientific Volume Imaging, Hilversum, Netherlands) and again run through PlantSeg with a minimum cell size of 300,000 voxels. If this did not improve the cell boundary predictions, images were excluded from the analysis. Cell volume was extracted from PlantSeg output tiff files using a threshold function and the Voxel counter plugin in ImageJ. 3D projections in Supplementary Movie 2 were created with Imaris software (Bitplane) version 9.7.0.

**DAPI staining and imaging of gametophore leaves**. We collected 8–9 gametophores per line after 3 weeks of culture on KNOP medium supplemented with microelements[49] and 5 mM ammonium tartrate. Gametophores were fixed for 1.5 h in 4% paraformaldehyde in 1 × PBS buffer, washed twice in 1 × PBS, incubated for 30 min in 4% driselase in 1 × PBS buffer, washed twice in 1 × PBS, incubated for 30 min in 0.1% Tween in 1 × PBS buffer, washed twice in 1 × PBS and incubated in 4′,6-diamidino-2-phenylindole (DAPI) solution (0.01 mg/L DAPI, 1.07 g/L MgCl₂·6H₂O, 5 g/L NaCl, 21.11 g/L TRIS in 1 mL Triton) overnight at 4 °C. Individual leaves (typically 3–6 leaves from each gametophore) were separated from the gametophore stem using a razor blade and mounted in 1 × PBS buffer for imaging. Images were acquired using a Zeiss Axioplan2 microscope (5 × 0.15-NA lens; Carl Zeiss Microscopy, Germany) equipped with a digital camera (Axiocam Mrc5, Carl Zeiss) and operated with AxioVision software. Imaging data were analyzed using ImageJ software. Gametophore and cell sizes were measured manually, and fluorescence intensity from nuclei stained with DAPI was measured using a "Threshold" function. Specialized conducting cells of the midrib and narrow cells of the leaf edge were excluded from cell area measurements, and conductive cells were excluded from DAPI fluorescence intensity measurements. Prism (GraphPad, San Diego, CA, USA) was used to plot the graphs, and to perform statistical analyses.

**Analysis of protonema ploidy**. Moss protonema was homogenized in 5 mL of distilled water and cultured on BCDAT agar plates covered with cellophane for 6 days before measurement. Next, small portion of protonema was chopped with a blade in 4′,6-diamidino-2-phenylindole (DAPI) solution (0.01 mg/L DAPI, 1.07 g/L MgCl₂·6H₂O, 5 g/L NaCl, 21.11 g/L TRIS in 1 ml Triton) and filtered through a 30 µm sieve. Fluorescence of the nuclei was determined with a PAS flow cytometer (Partec, Munster, Germany) using a 100 W high-pressure mercury lamp[50].

**Sequence analysis**. We used MAFFT ver. 7.043 (https://mafft.cbrc.jp/alignment/software/) to align the amino acid sequences of the selected full-length proteins, and then manually revised them with MacClade ver. 4.08 OSX (www.macclade.org) to remove gaps. The Jones–Taylor–Thornton model was used to construct maximum-likelihood trees using MEGA5 software (www.megasoftware.net). Statistical support for internal branches by bootstrap analysis was obtained using 1000 replications. The gene sequence information discussed in this article is available under the following accession numbers in Phytozome (www.phytozome.net): AtTPX2 (AT1G03780.3); OsTPX2 (LOC_Os07g32390.1); PpTPX2-1 (Pp3c17_11160V3.1); PpTPX2-2 (Pp3c1_25950V3.1); PpTPX2-3 (Pp3c24_8590V3.2); PpTPX2-4 (Pp3c23_4540V3.1); PpTPX2-5 (Pp3c5_10270V3.1); MpTPX2-1 (Mapoly0016s0083.1); MpTPX2-2 (Mapoly0105s0040.1) or in UNIPROT (www.uniprot.org): HsTPX2 (Q9ULW0); GgTPX2 (F1NW64); XlTPX2 (Q6NUF4).

**Reporting summary**. Further information on research design is available in the Nature Research Reporting Summary linked to this article.

## Data availability
The authors confirm that all relevant data supporting the findings of this study are available within the paper and its supplementary files. Plant lines used in this study are available upon request from corresponding authors (E.K. and G.G.).

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

## Acknowledgements

We thank Maya Hakozaki, Momoko Nishina, and Yuki Nakaoka for assistance with this project, Sebastian Hoernstein for help and advice on RNA extraction, Lennard Bohlender for help with qRT-PCR, Julian Knerr for providing vector with F-tractin for cloning, Raymundo Alfaro-Aco for comments on the TPX2 functional motifs, Peishan Yi and Mariana Costa for helpful comments on the manuscript, and the Life Imaging Center at the University of Freiburg for microscopic use. This work was funded by JSPS KAKENHI (17H06471, 22H02644, 22H04717), JSPS Joint Research Projects with UK Research and Innovation (to G.G.), and an Alexander von Humboldt Foundation Postdoctoral Fellowship (to E.K.). Additional support came from the German Research Foundation (DFG) under Germany's Excellence Strategy (CIBSS – EXC-2189 – Project ID 390939984 to R.R.).

## Author contributions

E.K. and G.G. designed the research project; G.G. and R.R supervised the project; E.K. and M.W.Y. performed experiments; E.K. analyzed the data; E.K., G.G., and R.R. wrote and edited the manuscript.

## Funding

## Competing interests

The authors declare no competing interests.
