## [Peer Review File · Nature Communications]

Spindle motility skews division site determination during asymmetric cell division in *Physcomitrella*REVIEWER COMMENTS

Reviewer #1 (Remarks to the Author):

The manuscript by Kozgunova et al characterizes the function of TPX2 proteins during cell division in *Physcomitrella*. Five TPX2 genes are identified, subcellular localization determined, and mutant analysis performed. Missense mutations in TPX2 1 to 4 (termed TPX2 1-4delta) produced only mild effects with elongated prophase nuclei and prolonged mitosis (NEB to anaphase). Since attempts at knocking out all five TPX2-5 were unsuccessful (presumably due to lethality), -5 knock-down lines were generated in the TPX2 1-4delta background. These lines were termed TPX2-5 HM. They showed stunted gametophytes with small and misshapen leaves. The authors attribute this phenotype to the observed misplaced cell division plane defects in the TPX2-5 HM lines. The main emphasis of the study is that the faulty division plane placement is due not to the typical causes (aberrant phragmoplast trajectories and/or misplaced nuclei prior to cell division) but is instead due to a rapid basal migration of the mitotic spindle itself. As this migration did not occur in the presence of very low latrunculin A, the authors propose a "tug-of-war" between actin and microtubules, where actin pulls the spindle toward the basal side of the cell, and microtubules, by way of TPX2's, favor the apical side. The TPX2-5 HM lines show other occasional mitotic defects, including spindle collapse and failed chromosomal partitioning.

The paper is well-written and easy to read. The figures are also nice and generally clear, and the movies are very helpful. My main criticism is that there is little analysis of the actual mechanism behind the spindle motility phenotype. There is no further investigation into the precise mechanism of this tug-of-war, making it essentially an observational finding. Attempts to observe actin organization did not go beyond trying LifeAct as an actin reporter, which did not work because it caused excessive bundling. There are other methods to visualize actin such as using the more reliable reporter of the second actin binding Fimbrin2 (FP-ABD2), immunological approaches, as well as fluorescently tagged phalloidin. Analysis of the relationships between actin, microtubules, and TPX2's is essential for any further mechanistic insights and would greatly strengthen the study. Similarly, deeper investigation into the nature of the stunted leaves/plants is important. There appear to be far fewer cells in the HM leaves. Why? The claim is made that the stunted phenotype is due solely to division defects and not to cell expansion. However, it is very clear from the leaf image in figure S5D that the cells are mostly larger, show greater variation in size, and are very disorganized. Given that the leaves are smaller, but the cells are fewer but bigger, loss of cell divisions altogether seems a likely culprit, but there is no analysis of this. It seems that variations in ploidy levels resulting from aberrant cell divisions would explain this variation in cell size, but the authors rule this out via flow cytometry of the nuclei (although the profiles do look very different in the HM lines). However, given the fact that multinucleate cells are observed, a similar gene dosage effect on cell size as for ploidy changes should influence cell size, would it not? Linking the different spatial scales of cell division, cell shape and organization, to broader leaf and plant morphology will greatly strengthen the paper and make it relevant to a much broader audience.

What is the mechanism of the prophase nucleus elongation in the TPX2 1-4delta lines? Their expression seems to be very low at this stage (very clear from the movies), but then increases drastically during the spindle/phragmoplast stages. Nuclear shape is often an actin-dependent trait. Was this affected in the latrunculin A experiments?

Being structurally distinct from TPX2 1-4delta, TPX2-5 is the only one that localizes not to the spindle poles, but instead to the center, just adjacent to the chromosomes, and follows them outward during anaphase as well. Are these specifically kinetochore microtubules? This important detail is glossed over in the results text and would stand a more thorough description. Some speculation in the discussion as to how this localization could link to the observed spindle migration role would be very helpful.

The title is over-reaching by saying the mechanism occurs in moss (isn't it possible that other species don't use this mechanism?) and should be specific to say *Physcomitrella patens* (unfortunately now changed by taxonomists to *Physcomitrium patens*). Similarly, the title reads as if the position of the mitotic spindle is the only mechanism of division plane determination, although any perturbation of actin and/or microtubules can cause mispositioning by other mechanisms.

Further comments, questions and recommendations are below. For brevity, they are not fully exhaustive in nature regarding all the finest details.

The paper would benefit greatly from some reorganization and presentation of the data. There is too much supplementary data, and most of it is actually primary data and should be in the main text. For example, Figure S5D should be in the main text as it nicely shows the obvious division plane defects in leaves of the HM lines. Some of the supplemental figures are not even described in the results section (such as S2 and S8). So, are they really necessary? Some of the images shown in the main figures are re-shown in the supplemental data (e.g. figure S2 and figure 1; figure S4 and figure 2); which is not technically forbidden, and it is mentioned in the supplemental figure legend; but it does give the impression of insufficient data. There is no information on experimental replicates, and the n-values for the latrunculin A experiments in figure 3E-F in particular are very low (for example n=2 for the 50nM treatment). Some controls are missing, for example figures 3E-G, and no WT control for the spindle tracking data in 3d.

It would be much clearer to put the TPX2-5 RNAi data along with the HM data, as they are essentially both knock-down experiments.

Due to the large increases in TPX2 expression during mitosis the signals of the FP-TPX2's and FP-Tubulin are very faint during prophase, and then over-saturated by cytokinesis. I think it would be fair to adjust the contrast for presentation in the figures. In fact, the dramatic increase in expression of the TPX2's during division in my opinion warrants some more emphasis in the text and figures, because this fact really jumps out when the movies are viewed (and most people don't look at supplemental movies unfortunately unless they are directly in the results section). Similarly, the blaring chloroplast autofluorescence in the green channels appears to be unavoidable with the hardware used for imaging, but the little asterisks on the images are confusing as it brings the attention to them instead of the true localization of the proteins. They could be removed and instead point to the actual signal of the FP reporters.

It would be much better to use different florescent protein colors for chromosomes and microtubules in figure 3 because a lot of the microtubule structure is masked by the bright chromosomal signal.

The methods section needs more details in a few places. For example, "Transformation was performed using a standard polyethylene glycol-mediated method using protoplasts" is not sufficient for other researchers to repeat the study. Effort should be made not to simply cite previous publications for a method. How was spindle area determined? Minor comment, while the results text says microtubule numbers were decreased in some mutant spindles, but since numbers were not actually counted, only fluorescence intensity measured, this should be made clear.

Does low-level LatA treatment rescue/mitigate the stunted and disorganized leaves in the HM line?

For the discussion it would be nice to speculate on why TPX2-5 appears to function in division plane placement during nuclear division but not cytokinesis, which seems odd given its huge expression and localization to the phragmoplast during cytokinesis. Why do the HM lines have slanted cross walls in caulonemal cells (Figure S8, not mentioned in text)? Is this due also to defective spindle positioning? It would seem to be something else. Is the spindle positioning mechanism present only in the gametophore initial cells during asymmetric division?

I hope this feedback will be helpful to the authors.

Reviewer #2 (Remarks to the Author):

This manuscript provides a detail characterization of the moss TPX2 and demonstrates the importance of spindle motility for cell division plane placement. The potential contribution of actin during this process is also very intriguing. It is stated that expression of mCherry-Lifeact rescued the spindle motility defects in TPX2 mutants, suggesting altered actin dynamic when mCherry-Lifeact is expressed. However, it might still be worthwhile to show actin localization labeled by Lifeact in WT and TPX2 mutants, since in the literatures the presence of actin was mostly observed in phragmoplast and not in spindle. If actin indeed contributes to spindle motility, the localization of actin filaments during early stage of cell division might reveal more information. A clean knockout of TPX2-5 can't be generated by homologous recombination indicates TPX2-5 is essential. It is curious why CRISPR was not utilized to generated TPX2-5 mutant alleles? Often time CRISPR could generates in frame deletion that significantly disrupts function of an essential gene but still produces viable plants. Imaging of the endogenously tagged TPX2 genes, and the gametophore cell division are nicely done. The spindle motility phenotype is very well quantified. These data and findings would be valuable to the field.

REVIEWER COMMENTS

Reviewer #1 (Remarks to the Author):

The manuscript by Kozgunova et al characterizes the function of TPX2 proteins during cell division in Physcomitrella. Five TPX2 genes are identified, subcellular localization determined, and mutant analysis performed. Missense mutations in TPX2 1 to 4 (termed TPX2 1-4delta) produced only mild effects with elongated prophase nuclei and prolonged mitosis (NEB to anaphase). Since attempts at knocking out all five TPX2-5 were unsuccessful (presumably due to lethality), -5 knock-down lines were generated in the TPX2 1-4delta background. These lines were termed TPX2-5 HM. They showed stunted gametophytes with small and misshapen leaves. The authors attribute this phenotype to the observed misplaced cell division plane defects in the TPX2-5 HM lines. The main emphasis of the study is that the faulty division plane placement is due not to the typical causes (aberrant phragmoplast trajectories and/or misplaced nuclei prior to cell division) but is instead due to a rapid basal migration of the mitotic spindle itself. As this migration did not occur in the presence of very low latrunculin A, the authors propose a “tug-of-war” between actin and microtubules, where actin pulls the spindle toward the basal side of the cell, and microtubules, by way of TPX2’s, favor the apical side. The TPX2-5 HM lines show other occasional mitotic defects, including spindle collapse and failed chromosomal partitioning.

The paper is well-written and easy to read. The figures are also nice and generally clear, and the movies are very helpful.

My main criticism is that there is little analysis of the actual mechanism behind the spindle motility phenotype. There is no further investigation into the precise mechanism of this tug-of-war, making it essentially an observational finding. Attempts to observe actin organization did not go beyond trying LifeAct as an actin reporter, which did not work because it caused excessive bundling. There are other methods to visualize actin such as using the more reliable reporter of the second actin binding Fimbrin2 (FP-ABD2), immunological approaches, as well as fluorescently tagged phalloidin. Analysis of the relationships between actin, microtubules, and TPX2’s is essential for any further mechanistic insights and would greatly strengthen the study.

To address this point, we generated transgenic moss lines expressing three possible F-actin reporters: F-tractin, Utrophin, and the actin-binding domain of the Fimbrin2 homologue identified in the Physcomitrella genome through homology search (Pp3c15_21790). Utrophin and PpFimbrin2 were deemed unsuitable for the visualization of actin, as they showed aggregates or preferential binding to actin in certain parts of the cell, respectively. In contrast, Citrine-F-tractin stained F-actin in a manner similar to that of Lifeact and, unlike Lifeact, did not affect the spindle motility phenotype in the *TPX2-HM1* mutant. Therefore, Citrine-F-tractin was used to observe actin distribution in dividing cells in the gametophore initial.

In both the control and *TPX2-5 HM1* lines, actin cables were distributed evenly around the cell cortex and showed clear filamentous structures. In contrast, actin was observed in the form of a “cloud” around the spindle, without visible structured elements (Fig. 6, n = 7 control, n = 13 *TPX2-5 HM1*). The cloud was not restricted to the spindle-proximal area but spread to the cortical region at all time frames during prometaphase and metaphase. The actin cloud was translocated with the motile spindle in the mutant (Fig. 6). These results further support the conclusion that actin is responsible for spindle motility.

We thank the reviewer for suggesting alternative actin makers.

Similarly, deeper investigation into the nature of the stunted leaves/plants is important. There appear to be far fewer cells in the HM leaves. Why? The claim is made that the stunted phenotype is due solely to division defects and not to cell expansion. However, it is very clear from the leaf image in figure S5D that the cells are mostly larger, show greater variation in size, and are very disorganized. Given that the leaves are smaller, but the cells are fewer but bigger, loss of cell divisions altogether seems a likely culprit, but there is no analysis of this. It seems that variations in ploidy levels resulting from aberrant cell divisions would explain this variation in cell size, but the authors rule this out via flow cytometry of the nuclei (although the profiles do look very different in the HM lines). However, given the fact that multinucleate cells are observed, a similar gene dosage effect on cell size as for ploidy changes should influence cell size, would it not? Linking the different spatial scales of cell division, cell shape and organization, to broader leaf and plant morphology will greatly strengthen the paper and make it relevant to a much broader audience.

We have conducted a more detailed analysis of the gametophore phenotype, including leaf size, cell size distribution, and ploidy.

First, many leaves in the *TPX2-5 HM1* line were severely deformed and/or underdeveloped (Fig. 3b). For example, in 14 out of 23 leaves, we could not visually distinguish specialized central conducting cells of the midrib or narrower cells on the edge of the leaf.

Second, we quantified and concluded that the leaves in the *TPX2-5HM1* line were smaller and had fewer cells compared to the GH (control) or *TPX2 1-4Δ* lines (Fig. 3c-e).

Third, the cell size was quantified. In *Physcomitrella*, the tip of the gametophore leaf constitutes a meristematic zone and consists of smaller cells, whereas the cells on the basal side of the leaf undergo expansion (Dennis et al. 2019). Consistent with this, the tip of the leaf contained smaller cells compared to the basal side of the leaf in both GH and *TPX2 1-4Δ* lines (Fig. 3f). In the *TPX2-5 HM1* line, although basal cells were comparable to the control in size, the cells at the tips were larger, skewing the characteristic size distribution (Fig. 3f). In addition, *TPX2-5 HM1* leaves occasionally had extremely large cells (3-4 times the normal size; arrow in Fig. 3b).

Finally, we evaluated ploidy by quantifying the fluorescence intensities of the DAPI-stained nuclei of each leaf cell. In control cells, ploidy was higher at the basal side, where larger cells predominated, suggesting that endoreduplication occurs in the basal region (Fig. 3e). In contrast, the tip cells also had higher DNA contents in the *TPX2-5 HM1* mutant. This phenotype corroborates the cell size abnormality in the mutant, as genome endoreduplication frequently correlates with an increase in cell size in plants. Please note ploidy analysis in Supplementary Fig. 7 were done using only protonemal cells and this data shows that protonema cells remain largely haploid in *TPX2-5* mutants.

Overall, these new phenotypic characterizations support our hypothesis that cumulative cell division defects are the main, if not the sole, contributor to the dwarf gametophore phenotype in *TPX2* mutants. For example, spindle motility would result in an abnormal cell size ratio of the daughter cells, which might affect cell cycle control, including DNA replication. Spindle collapse and the resultant cytokinesis failure, though only infrequently observed, would affect cell size and ploidy in individual cells that can affect the development of surrounding cells through changes in gene expression.

*What is the mechanism of the prophase nucleus elongation in the *TPX2 1-4delta* lines? Their expression seems to be very low at this stage (very clear from the movies), but then increases*

drastically during the spindle/phragmoplast stages. Nuclear shape is often an actin-dependent trait. Was this affected in the latrunculin A experiments?

We conducted live-cell imaging of *TPX2 1-4Δ* lines treated with 5 μ M LatA and did not detect any change in nuclear shape (Reviewer Figure, below). The lengths of prophase nuclei were also not significantly changed: $17.8 \pm 2.8 \mu\text{m}$ and $17.6 \pm 2.5 \mu\text{m}$ before and after LatA treatment, respectively (5 μ M, 10 min). We describe this experiment in the manuscript and report that the nuclear shape in this cell type does not depend on actin.

Reviewer Figure: Latrunculin A (5 μ M) treatment did not affect nuclear shape in the *TPX2 1-4Δ* mutant. Bar, 10 μ m

In the *Physcomitrella* protonema, nuclear deformation has been observed in a kinesin mutant (Yamada and Goshima. *Plant Cell*. 2018). Therefore, we speculate that the nuclear shape is controlled by the force generated by the perinuclear microtubule cytoskeleton and associated motors. Perinuclear microtubule accumulation during prophase at the apical side is reduced in the *TPX2 1-4Δ* line, which probably results in a force imbalance and nuclear elongation.

Being structurally distinct from TPX2 1-4delta, TPX2-5 is the only one that localizes not to the spindle poles, but instead to the center, just adjacent to the chromosomes, and follows them outward during anaphase as well. Are these specifically kinetochore microtubules? This important detail is glossed over in the results text and would stand a more thorough description. Some speculation in the discussion as to how this localization could link to the observed spindle migration role would be very helpful.

We thank the reviewer for highlighting this point. The mitotic localization of each TPX2 is slightly different from the others. TPX2-5 is particularly unique in that it is upregulated in anaphase. We have described this observation in more detail in the revised manuscript.

Polar enrichment was prominent throughout mitosis in TPX2-1, but not in others. Although the signal intensity significantly varied, TPX2-2 (low level), TPX2-4 (medium), and TPX2-5 (extremely low) decorated the entire spindle, except for the midzone, at metaphase. They probably bind to the kinetochore microtubules and interpolar microtubules that run in parallel, but not to the anti-parallel overlap of interpolar microtubules. The localization patterns were maintained in telophase for all TPX2s, except that the signal intensity was dramatically elevated for TPX2-5. During cytokinesis, TPX2-1 decorated the phragmoplast edge, whereas TPX2-2, TPX2-4, and TPX2-5 bound to the intra-phragmoplast microtubules that were aligned in parallel. TPX2-5 decoration persisted the longest during cytokinesis, while TPX2-2 and TPX2-4 gradually disappeared from late phragmoplast.

To illustrate these observations precisely, we have combined the localization data displayed in two Figures (former Fig. 1c and Supplementary Fig. 2) into one, and now present them as a main Figure 2. In the new Figure, the images with consistent brightness/contrast adjustment are presented for all TPX2s, which allows the comparison of the expression levels among paralogues or between the pre- and post-anaphase (Fig. 2). In addition, the brightness/contrast-enhanced images of low-expressed TPX2-5 are shown in a separate

panel, which clarifies the similarity and difference of the localization, compared to the highly expressed TPX2-1 and TPX2-4.

Because functionally redundant TPX2s show slightly different preferences in associated microtubules, the localization is not immediately associated with a spindle motility phenotype. Meanwhile, we found that actin, the driver of the spindle, surrounds the spindle and likely connects it to the cell cortex (Fig. 6, see Response to Comment #1). Thus, the smaller sized spindles and reduced MTs observed during prometaphase in the TPX2 mutant might not be resistant to the actin-driven force. This possibility is discussed in the revised manuscript.

The title is over-reaching by saying the mechanism occurs in moss (isn't it possible that other species don't use this mechanism?) and should be specific to say Physcomitrella patens (unfortunately now changed by taxonomists to Physcomitrium patens). Similarly, the title reads as if the position of the mitotic spindle is the only mechanism of division plane determination, although any perturbation of actin and/or microtubules can cause mispositioning by other mechanisms.

We have changed the title of the manuscript to “Spindle motility skews division site determination during asymmetric cell division in Physcomitrella”. It was suggested in the bryophyte community that Physcomitrella (non-italic) can be used as a common name for Physcomitrium patens, as Arabidopsis is often used for Arabidopsis thaliana.

Further comments, questions and recommendations are below. For brevity, they are not fully exhaustive in nature regarding all the finest details. The paper would benefit greatly from some reorganization and presentation of the data. There is too much supplementary data, and most of it is actually primary data and should be in the main text. For example, figure S5D should be in the main text as it nicely shows the obvious division plane defects in leaves of the HM lines. Some of the Supplementary figures are not even described in the results section (such as S2 and S8). So, are they really necessary?

Following this advice, we have reorganized the manuscript. For example, the former Supplementary Fig. 1 and 6 were moved to the main text as Fig. 2 and 4, respectively. Supplementary Fig. 2 and 7 have been removed from the revised manuscript. Modified Fig. 3b-g encompasses a series of analyses on the leaf phenotype. The discussion section was also reorganized to enhance readability.

Some of the images shown in the main figures are re-shown in the Supplementary data (e.g. figure S2 and figure 1; figure S4 and figure 2); which is not technically forbidden, and it is mentioned in the Supplementary figure legend; but it does give the impression of insufficient data.

We appreciate the reviewer's comment. After manuscript reorganization, there are no more overlapping data between the main and supplementary figures.

There is no information on experimental replicates, and the n-values for the latrunculin A experiments in figure 3E-F in particular are very low (for example n=2 for the 50nM treatment).

We have added the information on experimental replicates to the Methods. In brief, imaging was conducted at least twice to determine the TPX2 localization, whereas three or more independent experiments were performed for phenotype observation and analyses.

Regarding the low n values in Fig. 5e-f (former Fig. 3E-F), we regret that the provided values were misleading as more cells were observed. For example, we observed 18 gametophore initials for LatA 50nM treatment, but only 2 out of the 18 showed spindle motility. The main message in this experiment was that 16 out of 18 cells no longer showed spindle motility even

with this low dosage of LatA. Therefore, in the revised manuscript, we stated this information in the main text and removed the data on speed and travel distance from the graph for low-dosage LatA treatments.

Some controls are missing, for example figures 3E-G, and no WT control for the spindle tracking data in 3d.

The tracks in Fig. 5d, and graphs in Fig. 5e-g (former Fig. 3D–G) show how spindle motility was affected under various conditions (e.g., in the presence of the MT-stabilizing taxol). Since basal spindle motility was never observed in wild-type and *TPX2 1-4Δ* lines, no data could be presented in these figures.

As a control for these experiments, we analyzed a *TPX2-5 HM1* line treated with DMSO, which was used as a solvent for all inhibitors. This is stated in the legend (Fig. 5g) in the revised manuscript.

It would be much clearer to put the TPX2-5 RNAi data along with the HM data, as they are essentially both knock-down experiments.

We agree that both are essentially knock-down experiments. Although we tried to reorganize the Results section following this suggestion, we felt that the current order works better, as different cell types/life stages were analyzed for RNAi and *HM* lines. For *TPX2-5* RNAi lines, we analyzed only cell division in the protonema, since RNAi induction caused severe growth defects and we could not image the gametophores. Conversely, the phenotype in protonema of *TPX2-5 HM* lines was so mild that the gametophore development was mostly analyzed.

Due to the large increases in TPX2 expression during mitosis the signals of the FP-TPX2's and FP-Tubulin are very faint during prophase, and then over-saturated by cytokinesis. I think it would be fair to adjust the contrast for presentation in the figures.

In fact, the dramatic increase in expression of the TPX2's during division in my opinion warrants some more emphasis in the text and figures, because this fact really jumps out when the movies are viewed (and most people don't look at Supplementary movies unfortunately unless they are directly in the results section).

We agree that in some cases, particularly for *TPX2-5*, the signals become saturated during cytokinesis due to very strong localization of phragmoplast microtubules. In the revised version of the manuscript, we prepared two panels for *TPX2-5* (Fig. 2e), one with adjusted brightness/contrast that allows visualization of the localization to the spindle and one without adjustments. We also described *TPX2* localization in greater detail.

Similarly, the blaring chloroplast autofluorescence in the green channels appears to be unavoidable with the hardware used for imaging, but the little asterisks on the images are confusing as it brings the attention to them instead of the true localization of the proteins. They could be removed and instead point to the actual signal of the FP reporters.

We removed asterisks indicating chloroplast autofluorescence and added arrowheads indicating the localization of the proteins tagged with fluorophores.

It would be much better to use different florescent protein colors for chromosomes and microtubules in figure 3 because a lot of the microtubule structure is masked by the bright chromosomal signal.

Unfortunately, in the lines used for imaging in Fig. 5 (former Fig. 3), microtubules and chromosomes were both labelled with red fluorophores, mCherry and RFP, respectively. Therefore, both structures were imaged with a single channel and could not be visualized

using different colors. The fluorophore selection was made at the time due to technical reasons, such as the antibiotics available for line selection or saving the green channel for possible future use. Retrospectively, however, the availability of the green channel was helpful in further visualizing actin in the mutant with Citrine-F-Tractin (Fig. 6).

The methods section needs more details in a few places. For example, "Transformation was performed using a standard polyethylene glycol-mediated method using protoplasts" is not sufficient for other researchers to repeat the study. Effort should be made not to simply cite previous publications for a method.

We have described our experimental procedures in greater detail in the revised manuscript.

How was spindle area determined?

The spindle area was measured based on max intensity Z-projection images of metaphase, where the spindle border was tracked manually. This information has been added to the figure legend (Fig. 5c).

Minor comment, while the results text says microtubule numbers were decreased in some mutant spindles, but since numbers were not actually counted, only fluorescence intensity measured, this should be made clear.

We have rephrased the corresponding sentences.

Does low-level LatA treatment rescue/mitigate the stunted and disorganized leaves in the HM line?

To address this, we supplied a range of LatA concentrations from 50 pM to 100 nM to the culture medium onto which moss protonema tissue was inoculated. In either condition, the stunted gametophore phenotype was not rescued, although higher LatA concentrations inhibit protonema growth and more gametophores become visible in hypomorphic mutants (Supplementary Fig. 5).

For the discussion it would be nice to speculate on why TPX2-5 appears to function in division plane placement during nuclear division but not cytokinesis, which seems odd given its huge expression and localization to the phragmoplast during cytokinesis.

In general, it is not trivial to identify the post-anaphase function of a gene essential for metaphase spindle formation, because telophase and cytokinesis do not proceed normally if there is severe deformation of the metaphase spindle. This was the case with PpTPX2: the most severe phenotype upon RNAi was the spindle collapse and multipolar formation, and it is not clear whether the abnormal phragmoplast structure observed in the RNAi line is due to the lack of a specific function of TPX2 during telophase or the consequence of pre-anaphase defects. Previously, we concluded the function of microtubule amplifier (augmin, HURP) during cytokinesis in human tissue culture cells, by developing a sophisticated mitotic synchronization assay combined with fine-tuned depletion of target proteins (Uehara and Goshima. J Cell Biol. 2010). Presently, this type of assay cannot be conducted technically in moss protonema. In the revised discussion, we have stated that there could be post-anaphase functions of TPX2 and that we could not identify them in the current study.

Why do the HM lines have slanted cross walls in caulonemal cells (Fig. S8, not mentioned in text)? Is this due also to defective spindle positioning? It would seem to be something else.

According to our observations, there was no spindle motility in the protonema of any type of cell in the *TPX2-5 HM1* line, including branching cells that also undergo asymmetric cell division. It is still unknown what determines the angle of cross walls in *Physcomitrella*, and so

the mechanism underlying this phenotype is unclear; possibly microtubule dynamics were altered to a certain extent in the phragmoplast, leading to mild defects in phragmoplast guidance.

Is the spindle positioning mechanism present only in the gametophore initial cells during asymmetric division?

With currently available mutants, we can confidently conclude the presence of the spindle positioning mechanism only in the gametophore initial. We also observed positional instability of the spindle in protonema cells in the *TPX2-5* RNAi lines, such as spindle rotation and movement (Fig. 7d, Supplementary movie 6). This suggests that the spindle positioning mechanism also operates in protonema cells to a certain extent. However, spindle movements in RNAi lines are usually observed together with other phenotypes, such as chromosome mis-segregation or spindle collapse, complicating further analysis and interpretation. This point has been discussed in the revised manuscript.

Reviewer #2 (Remarks to the Author):

This manuscript provides a detail characterization of the moss TPX2 and demonstrates the importance of spindle motility for cell division plane placement. The potential contribution of actin during this process is also very intriguing.

1. It is stated that expression of mCherry-Lifeact rescued the spindle motility defects in TPX2 mutants, suggesting altered actin dynamic when mCherry-Lifeact is expressed. However, it might still be worthwhile to show actin localization labeled by Lifeact in WT and TPX2 mutants, since in the literatures the presence of actin was mostly observed in phragmoplast and not in spindle. If actin indeed contributes to spindle motility, the localization of actin filaments during early stage of cell division might reveal more information.

We tried several alternative actin probes and discovered that Citrine-F-tractin did not suppress spindle motility in the mutant. Thus, we used Citrine-F-tractin to visualize actin during the first division of gametophore initial. Cortical actin was observed in cables and filaments and was distributed evenly around the cells in both the control and mutant lines. We observed an actin cloud around the spindle in both the control and *TPX2-5* mutant lines, which moved together with the spindle. We believe this observation further supports our conclusion that actin is the main driving force behind spindle translocation. These new results have been added to the revised manuscript (Fig. 6).

2. A clean knockout of TPX2-5 can't be generated by homologous recombination indicates TPX2-5 is essential. It is curious why CRISPR was not utilized to generated TPX2-5 mutant alleles? Often time CRISPR could generates in frame deletion that significantly disrupts function of an essential gene but still produces viable plants.

We would like to thank the reviewer for suggesting this experiment. First, we introduced a frameshift of *TPX2-5* in a wild-type background. However, the frameshift line we obtained did not have any noticeable phenotype (this data was not included). We speculated that this could be due to the expression from other start codons present in *the TPX2-5* gene. Therefore, we tried to introduce a large deletion with CRISPR by targeting the first and fifth exons of the *TPX2-5* gene. Two independent lines were isolated, in which a large part of the *TPX2-5* coding sequence was deleted. We characterized these lines and did not detect any noticeable phenotype in the protonema or gametophores (Supplementary Fig. 4). Based on these results, we have revised our initial suggestion that *TPX2-5* is an essential gene in the wild-type background.

REVIEWER COMMENTS

Reviewer #1 (Remarks to the Author):

I would like to thank the authors for their diligent work and additional experiments. The authors have addressed most of my comments in the revised manuscript. I have some questions and comments mostly regarding the new actin data and the overall interpretation.

An actin marker line was successfully generated using the actin marker F-tractin, and is shown in figure 6. In both wild type and tpx2-5 HM1, a cortical actin network is observed, whereas the internal cytoplasm largely lacks filaments and is enriched around the nucleus throughout cell division in both genotypes. This internal fluorescent signal is described as an actin cloud (or cytoplasmic actin in the figure 6 title). Given that the fluorescence comes from unbound FP-tractin in the cytoplasm, I would suggest changing the terminology to more accurately reflect this (i.e. the cytoplasm).

As far as I can tell from the authors description and images, there is no change in the organization of the cortical actin in the mutant (although only one cell for each genotype is shown). In this cell, as the spindle migrates toward the base in the mutant, the cytoplasm moves along with it. Is there any change in the cortical actin surrounding the cytoplasm? Based on the images in figure 6, it appears that the mutant vacuoles are not concentrated basally (as in control), but instead are more broadly distributed, even prior to NEBD. Along with the vacuolar morphology defects, the cytoplasm also extends farther toward the base of the cell, and again even occurring prior to division. The midplane images of HM1 have what looks like to me an increase in cytoplasmic (i.e. non-cortical) filaments near the apical tip of the cell. Only one cell is shown, so it's unclear as to how much variability is seen, but presumably this is a representative image. No quantification is presented. It's not clear why the image of chlorophyll autofluorescence is shown in figure 6, particularly when there is none in the f-tractin lines. The figure title says cytoplasmic actin moves with the spindle in the HM1 line, but shouldn't it be that the cytoplasm moves along with the spindle? Is the cytoplasmic migration causal of the spindle motility, or just a passive movement, following the spindle? Based on what is shown and described, we are not gaining much mechanistic insight regarding the role of actin in spindle positioning.

The new images of the mutant leaves and extensive quantification of cell numbers and sizes are very helpful (figure 3). An increase in cell size near the tip of the leaves, and decrease in cell numbers is presented along with new data addressing ploidy levels based on DAPI staining. The DAPI data presents nuclear fluorescence intensity measurements and shows that along with the developmental gradient in cell size, where older, larger cells are at the base, the large cells near the base also have higher DNA content, indicating endoreduplication. In the HM1 mutant the developmental gradient from tip to base is lost, and cells are larger at the apical side of the leaf. Along with this, the enlarged cells near the apex have higher DNA content. Whether the cells are larger because of the increased ploidy, or vice versa is not known.

Regarding the previous analysis of DNA content in protonema (Fig S7, formerly S9), the figure is no longer described as flow cytometry, but rather as DAPI fluorescence. And all mentions of flow cytometry are in the manuscript have been removed, (once in results, once in methods (a full section for it), and once in acknowledgements). Perhaps the authors can clarify this.

I have a few questions regarding the interpretation of data regarding the effect of the misplaced cell divisions in the mutant. Clearly, the first asymmetric division occurs towards the base instead of the tip of the cell in the mutant, and clearly the leaves are stunted and misshapen. Are the authors saying that this first division problem is responsible for the whole of the subsequent leaf development? In other words, the subsequent cell division planes were all normal in the developing leaf? Being symmetric divisions, there would presumably not be spindle motility (as with protonemal cells), but are there problems in division plane orientations overall? Given the planar nature of the leaf, division plane orientations would presumably be tightly controlled during the establishment of overall leaf

shape. There is a brief mention in the results regarding later cell divisions in the gametophore (line 196): "In six cases, spindle motility or rotation was observed in the second or later divisions, while the first cell division site appeared normal (Supplementary Movie 4)." Can the authors clarify? Six out of how many observed post-initial divisions?

Fig 3g has a typo in GH graph. Can the authors present this data as frequency instead of counts? This would be particularly helpful, as there appears to be fewer cells counted in HM1. It may also help to normalize the X-axis to relative intensity for comparison across the three genotypes.

In the abstract, it would benefit from mentioning the fact that this division mechanism does not rely on a PPB. Also, in line 29, it reads as if spindle motility is a normal process, but the mutant has problems with this normal process. Just suggestions to help, as these concepts don't really come through in the abstract.

Some of the figure panels are called out of order, and figure 5 is split into two results sections. Usually journals require calling panels in order. Just a mention.

Discussion, line 322 says the *tpx2* mutant has cell sizes comparable to controls in gametophores. Is this from the first version and not changed, or am I missing something?

Discussion, Line 332 says spindle collapse and cytokinesis failure in 20% of gametophore cells of the mutant. In figure 5b, the HM mutant has spindle collapse in 2 of 19 cells. Does this relate to the RNAi lines in protonemal cells also?. Clarification?

Discussion line 370, and in the model figure 8: taxol's buffering of the spindle motility phenotype makes sense. Perhaps the authors could explain the less obvious observation that oryzalin does also?

Reviewer #2 (Remarks to the Author):

The revised manuscript is clear and has addressed most of the questions raised by reviewers.

Reviewer #1 (Remarks to the Author):

I would like to thank the authors for their diligent work and additional experiments. The authors have addressed most of my comments in the revised manuscript. I have some questions and comments mostly regarding the new actin data and the overall interpretation.

We would like to thank the reviewer for taking an interest in our study and providing detailed comments. We have conducted an additional analysis on actin distribution and made the suggested changes to the manuscript and figures, as described in our point-by-point responses below.

1. An actin marker line was successfully generated using the actin marker F-tractin, and is shown in figure 6. In both wild type and tpx2-5 HM1, a cortical actin network is observed, whereas the internal cytoplasm largely lacks filaments and is enriched around the nucleus throughout cell division in both genotypes. This internal fluorescent signal is described as an actin cloud (or cytoplasmic actin in the figure 6 title). Given that the fluorescence comes from unbound FP-tractin in the cytoplasm, I would suggest changing the terminology to more accurately reflect this (i.e. the cytoplasm).

Our interpretation of the data was that the fluorescent signals of FP-tractin in the cytoplasm are a combination of signals coming from unbound FP-tractin and FP-tractin bound to actin filaments. Actin filaments in the cytoplasm are likely short and thin and are thus not clearly discernable. In fact, we occasionally observed filamentous signals and brighter speckles in the cytoplasm, suggesting that not all cytoplasmic signals come from unbound FP-tractin (indicated by arrowheads in the new Fig. 6).

The term “actin cloud” comes from the field of meiosis in mammals, where it is used to describe the amorphous actin meshwork that plays an important role in the repositioning of acentrosomal meiotic spindles. Individual actin filaments in the cytoplasm are also hardly discernable in this system; nonetheless, spindle motility is effectively suppressed by an actin inhibitor (Azoury et al., *Cur Biol* 2008), and several actin-dependent models have been proposed on the mechanism of spindle motility (Yi et al., *JCB* 2013; Holubcová et al., *Nat Cell Biol* 2013). We agree that describing the internal fluorescent signal observed in gametophore initials as an “actin cloud” was misleading. Following the reviewer’s advice, we have more carefully described the F-tractin signals, replacing the term “actin cloud” to “cytoplasmic actin”.

2. As far as I can tell from the authors description and images, there is no change in the organization of the cortical actin in the mutant (although only one cell for each genotype is shown). In this cell, as the spindle migrates toward the base in the mutant, the cytoplasm moves along with it. Is there any change in the cortical actin surrounding the cytoplasm? Based on the images in figure 6, it appears that the mutant vacuoles are not concentrated basally (as in control), but instead are more broadly distributed, even prior to NEBD. Along with the vacuolar morphology defects, the cytoplasm also extends farther toward the base of the cell, and again even occurring prior to division. The midplane images of HM1 have what looks like to me an increase in cytoplasmic (i.e. non-cortical) filaments near the apical tip of the cell. Only one cell is shown, so it’s unclear as to how much variability is seen, but presumably this is a representative image. No quantification is presented.

[Cortical actin] We could not identify a difference in cortical actin between control and TPX2-5 mutant lines. This is not surprising given that cell growth was not affected in the TPX2-5 mutant; abnormal actin dynamics would likely result in cell growth retardation.

[Vacuole] We thank the reviewer for pointing out the apparent difference in vacuole morphology and distribution. Indeed, they look considerably different between control and

mutant lines in the current “single frame” image panels. However, the apparent difference was largely an artifact of z-plane selection; vacuole morphology and distribution are markedly different if we select different z-planes (new Fig. 6a). Overall, we could not detect a dramatic difference between samples; large vacuoles occupied the basal region in both control and mutant lines in early mitosis, as indicated in the z-projection image. In the newly revised figure, we have displayed three panels, encompassing the cortical and middle sections, and omitted the outline highlighting the vacuole. We have also made a new Supplemental Movie 5, showing three cells for each genotype.

[Quantification] The reviewer raised an interesting point that a change in actin distribution might underlie spindle motility. To test this hypothesis, we measured Citrine-F-tractin fluorescence intensity across the z-stack in the area adjacent to the apical and basal spindle poles. Specifically, we created a square region of a fixed size (8 x 4 μm) and placed it at a) the basal spindle pole and b) the apical spindle pole. We measured mean gray values in the Citrine channel for each of z-slices and summed them. The fluorescence intensity data are described in Fig. 6b (control, n = 12 timepoints of 6 cells; *TPX2-5 HM1* mutant, n= 17 timepoints of 8 cells). The relative intensity (basal/apical) was not significantly different from 1 (two-tailed unpaired t-test, p = 0.65 [control] and 0.07 [mutant]). Thus, actin distribution is unlikely to be biased at either pole during spindle motility (mutant) or at the steady state (control).

3. It's not clear why the image of chlorophyll autofluorescence is shown in figure 6, particularly when there is none in the f-tractin lines. The figure title says cytoplasmic actin moves with the spindle in the HM1 line, but shouldn't it be that the cytoplasm moves along with the spindle? Is the cytoplasmic migration causal of the spindle motility, or just a passive movement, following the spindle? Based on what is shown and described, we are not gaining much mechanistic insight regarding the role of actin in spindle positioning.

[Chloroplast autofluorescence] To show that the fluorescent signals are actually derived from fluorescent protein rather than autofluorescence of chloroplasts or unknown sources, we included a control image, which was acquired with an identical microscopy setup but using a line without fluorescent protein expression. In the control image, autofluorescent chloroplasts, but not the cytoplasm, produced signals; this provides confidence that the cytoplasmic signals in the transgenic line derived from Citrine-F-tractin were not autofluorescence. The reason why chloroplasts were not clearly visible in the transgenic line was that the Citrine-F-tractin signal was much brighter than chloroplast autofluorescence. In the newly revised manuscript, we have omitted the control image, as we realized that it was more confusing to present the control in this case.

[Actin dynamics and mechanism of spindle motility] Following the reviewer's advice, the figure title has been revised to “Actin is detected in the cytoplasm and cortex during spindle motility in the *TPX2-5 HM1* line”. The unbiased localization of F-tractin during spindle motility reinforces the idea that actin persistently applies force on the spindle, and this force is counteracted by microtubules in wild-type. In contrast, with diminished microtubules in the mutant, as suggested by a smaller spindle size (Fig. 5c), actin-dependent force is the dominant force translocating the spindle. We agree that the exact mechanism remains unclear in the current study; specifically, we did not explore which motor(s) and/or actin regulators are responsible for generating force. However, this study demonstrates that the plant spindle is motile and that actin is involved in the motility (based on latrunculin treatment); these are new findings that we believe will be of great interest to the broad readership of *Nature Communications*.

[Cytoplasmic migration] Whether actin-dependent cytoplasmic migration (i.e., “flow”) is causal in the spindle motility or simply represents passive movement following the spindle was not formally tested. Because cytoplasmic streaming has not been detected in mosses (Pressel et al., Ann Bot 2008), we believe that the former is less likely to occur. We have added this notion to the newly revised manuscript.

4. The new images of the mutant leaves and extensive quantification of cell numbers and sizes are very helpful (figure 3). An increase in cell size near the tip of the leaves, and decrease in cell numbers is presented along with new data addressing ploidy levels based on DAPI staining. The DAPI data presents nuclear fluorescence intensity measurements and shows that along with the developmental gradient in cell size, where older, larger cells are at the base, the large cells near the base also have higher DNA content, indicating endoreduplication. In the HM1 mutant the developmental gradient from tip to base is lost, and cells are larger at the apical side of the leaf. Along with this, the enlarged cells near the apex have higher DNA content. Whether the cells are larger because of the increased ploidy, or vice versa is not known.

The reviewer has accurately summarized our experimental data. Regarding the last point, the generally accepted theory in plants is that increased polyploidy causes cell enlargement (Kondorosi et al., Curr Opin Plant Biol. 2000; Robinson et al., The Plant Cell 2018). We are unaware of the reverse relationship, in which cell enlargement triggers DNA endoreduplication.

5. Regarding the previous analysis of DNA content in protonema (Fig S7, formerly S9), the figure is no longer described as flow cytometry, but rather as DAPI fluorescence. And all mentions of flow cytometry are in the manuscript have been removed, (once in results, once in methods (a full section for it), and once in acknowledgements). Perhaps the authors can clarify this.

In the revised manuscript, we included analyses of DAPI signals in gametophores, measured manually with ImageJ, and in protonemal cells, measured using a PAS cell analyzer. Because both were analyses of ploidy contents, we chose to provide coherent titles in the figures and method sections.

6. I have a few questions regarding the interpretation of data regarding the effect of the misplaced cell divisions in the mutant. Clearly, the first asymmetric division occurs towards the base instead of the tip of the cell in the mutant, and clearly the leaves are stunted and misshapen. (Q1) Are the authors saying that this first division problem is responsible for the whole of the subsequent leaf development? In other words, the subsequent cell division planes were all normal in the developing leaf? (Q2) Being symmetric divisions, there would presumably not be spindle motility (as with protonemal cells), but are there problems in division plane orientations overall? Given the planar nature of the leaf, division plane orientations would presumably be tightly controlled during the establishment of overall leaf shape. (Q3) There is a brief mention in the results regarding later cell divisions in the gametophore (line 196): “In six cases, spindle motility or rotation was observed in the second or later divisions, while the first cell division site appeared normal (Supplementary Movie 4).” Can the authors clarify? Six out of how many observed post-initial divisions?

Multiple questions are included in this comment. We have numbered each question above (Q1–Q3) and provided answers below (A1–A3).

A1. We do not infer that this first division problem is responsible for the whole of the subsequent leaf development. Our data suggest that the gametophore phenotype is a cumulative result of defects in multiple cell divisions that occur during leaf development. First,

we observed normal first divisions without spindle motility in 3 of 19 cases, whereas we never observed a gametophore of normal morphology; this can be explained by division problems arising in later divisions. Second, we monitored 2- and 4-cell stages of the gametophore in which the first division plane was observed at a normal position and observed spindle motility in 6 of 10 cell divisions (Supplementary Movie 4). Finally, in 7 of 11 cases where the first cell division was defective, we observed spindle motility in the second and/or third divisions.

A2. We currently do not have the means to investigate the spindle motility phenotype or division plane orientation specifically during symmetric division in the mature gametophore. Mature gametophores were always abnormal in cell shape, size, and numbers, as well as overall morphology (Fig. 3b). However, given severe defects in early developmental stages, the phenotype of mature gametophores could not be readily interpreted.

A3. We apologize for not mentioning N for this observation. It was ten, as detailed in A1 above.

7. Fig 3g has a typo in GH graph. Can the authors present this data as frequency instead of counts? This would be particularly helpful, as there appears to be fewer cells counted in HM1. It may also help to normalize the X-axis to relative intensity for comparison across the three genotypes.

We thank the reviewer for noticing the typo. Following their advice, we have converted the y-axis label to “frequency” in Fig. 3g.

On the other hand, we do not think normalization of the x-axis is necessary because we applied the identical sample preparation and microscopy setup, such as laser power and exposure time, to all three samples, and the values (a.u.) were comparable between samples.

8. In the abstract, it would benefit from mentioning the fact that this division mechanism does not rely on a PPB. Also, in line 29, it reads as if spindle motility is a normal process, but the mutant has problems with this normal process. Just suggestions to help, as these concepts don't really come through in the abstract.

We have modified the abstract to highlight that asymmetric cell division in *Physcomitrella* does not rely on PPB and that spindle motility is not normally observed.

9. Some of the figure panels are called out of order, and figure 5 is split into two results sections. Usually journals require calling panels in order. Just a mention.

We have fixed the order problem; the figure panels, when first mentioned in the text, have been called in order. We do not think it is a problem that figure 5 is split into two results sections; however, we will edit the text and/or figure if the journal does not accept this format.

10. Discussion, line 322 says the tpx2 mutant has cell sizes comparable to controls in gametophores. Is this from the first version and not changed, or am I missing something?

We meant to describe that *TPX2-5 HM1* mutant cells were, on average, not smaller than the control, and therefore, the cell expansion defect was unlikely to be the cause of smaller gametophores. We have reformulated this sentence to avoid confusion.

11. Discussion, Line 332 says spindle collapse and cytokinesis failure in 20% of gametophore cells of the mutant. In figure 5b, the HM mutant has spindle collapse in 2 of 19 cells. Does this relate to the RNAi lines in protonemal cells also?. Clarification?

We thank the reviewer for pointing out our inconsistent descriptions. Spindle collapse in the gametophore initial occurred in 2 of 19 cells, as accurately described in Fig. 5b. Our description on line 332 (“~20%”) reflects our calculation error. We have corrected it to ~10%.

In the RNAi lines in protonemal cells, 3 of 52 cells (combined from both RNAi lines) had spindle collapse (Fig. 7d).

12. Discussion line 370, and in the model figure 8: taxol's buffering of the spindle motility phenotype makes sense. Perhaps the authors could explain the less obvious observation that oryzalin does also?

Indeed, the effect of oryzalin is more complex, and it is difficult to explain the observation at a mechanistic level. Overall, spindle motility phenotypes became more variable; for instance, in some cases, the spindle moved toward the apical side of the cell, which was never observed in the untreated *TPX2-5* mutant. In the newly revised manuscript, we have mentioned one possibility: oryzalin may further attenuate the interaction between actin and microtubules, which randomizes and slows spindle motility.

REVIEWERS' COMMENTS

Reviewer #1 (Remarks to the Author):

I would again like to thank the authors for their additional work and patience with my sometimes nit-picky comments. All of my comments have now been addressed. I should add that the ability of a study to stand up to a very tough review attests to its high quality, and this greatly strengthened paper is sure to have a large impact on the field. The powerful experimental system developed by the authors is commendable and will certainly continue to be a rich source of exciting discoveries.